# Plant-nanoparticles enhance anti-PD-L1 efficacy by shaping human commensal microbiota metabolites

Yun Teng [1,11] ✉, Chao Luo [1,2,11], Xiaolan Qiu[1,3,11], Jingyao Mu[1], Mukesh K. Sriwastva[1], Qingbo Xu[4], Minmin Liu[1,3], Xin Hu[5], Fangyi Xu[1], Lifeng Zhang[1], Juw Won Park [1,6], Jae Yeon Hwang [1], Maiying Kong [1,6], Zhanxu Liu[1], Xiang Zhang [7], Raobo Xu [7], Jun Yan[1], Michael L. Merchant[8], Craig J. McClain [9] & Huang-Ge Zhang [1,4,10] ✉

Diet has emerged as a key impact factor for gut microbiota function. However, the complexity of dietary components makes it difficult to predict specific outcomes. Here we investigate the impact of plant-derived nanoparticles (PNP) on gut microbiota and metabolites in context of cancer immunotherapy with the humanized gnotobiotic mouse model. Specifically, we show that ginger-derived exosome-like nanoparticle (GELN) preferentially taken up by *Lachnospiraceae* and *Lactobacillaceae* mediated by digalactosyldiacylglycerol (DGDG) and glycine, respectively. We further demonstrate that GELN aly-miR159a-3p enhances anti-PD-L1 therapy in melanoma by inhibiting the expression of recipient bacterial phospholipase C (PLC) and increases the accumulation of docosahexaenoic acid (DHA). An increased level of circulating DHA inhibits PD-L1 expression in tumor cells by binding the PD-L1 promoter and subsequently prevents c-myc-initiated transcription of PD-L1. Colonization of germ-free male mice with gut bacteria from anti-PD-L1 non-responding patients supplemented with DHA enhances the efficacy of anti-PD-L1 therapy compared to controls. Our findings reveal a previously unknown mechanistic impact of PNP on human tumor immunotherapy by modulating gut bacterial metabolic pathways.

Plant molecules encapsulated in nanoparticles (PNP), such as exosome-like nanoparticles (ELN) are present in a variety of foods[1,2] and play a critical role in modulating vital gut microbiota activity and host physiological systems, both locally and systemically[3]. Recent studies indicate that ginger-derived ELN (GELN) can alleviate colitis by modulating the tryptophan metabolism of gut *Lactobacillus* in the mouse[4], and garlic-ELNs are preferentially taken up by gut *Verrucomicrobia* phylum which contributes to reversing insulin resistance of

[1]Brown Cancer Center, University of Louisville School of Medicine, Louisville, USA. [2]Department of Central Laboratory, The affiliated Huai'an First People's Hospital of Nanjing Medical University, Huai'an, Jiangsu, China. [3]Department of Breast and Thyroid Surgery, The affiliated Huai'an First People's Hospital of Nanjing Medical University, Huai'an, Jiangsu, China. [4]Department of Microbiology and Immunology, University of Louisville, Louisville, KY, USA. [5]Department of Genomic Medicine, University of Texas MD Anderson Cancer Center, Houston, TX, USA. [6]Department of Bioinformatics and Biostatistics, SPHIS, University of Louisville, Louisville, KY, USA. [7]Department of Pharmacology and Toxicology, University of Louisville, Louisville, KY, USA. [8]Kidney Disease Program and Clinical Proteomics Center, University of Louisville, Louisville, KY, USA. [9]Department of Medicine, Division of Gastroenterology, Hepatology and Nutrition, University of Louisville School of Medicine, Louisville, KY, USA. [10]Robley Rex Veterans Affairs Medical Center, Louisville, KY, USA. [11]These authors contributed equally: Yun Teng, Chao Luo, Xiaolan Qiu. ✉e-mail: yun.teng@louisville.edu; h0zhan17@louisville.edu

diabetes[5]. Given that PNP has been identified from a variety of different types of edible plants, whether and how these PNP may be selectively taken up by certain species of human gut microbiota and further exert the diversity of modulation of bacterial metabolic products has yet to be elucidated.

Although it is well established that "eat me"/"don't eat me" signaling via phagocytic pathways is essential in maintaining host homeostasis in mammals[6,7], its presence in diet-derived PNP for bacteria remains understudied. Addressing "eat me"/"don't eat me" signals on the nanoparticles from the diet could emerge as groundbreaking options for future therapy by modulating gut microbiota homeostasis[5].

Tumors modulate the microenvironment to avoid the host immune response, such as upregulating checkpoint receptor ligands leading to the reduction of tumor-infiltrating lymphocytes and resistance to immunotherapy[8]. Despite the emerging importance of immunotherapy as a new treatment strategy for numerous advanced cancers, there is a disparity in the response of cancer patients to this regime[9]. Ongoing evidence suggests that the gut microbiome exerts a profound impact on the efficacy of cancer immunotherapy[10,11]. *Bacteroides* and *Lactobacillus* are proposed to benefit the anti-CTLA-4 and anti-PD-1 therapy in melanoma and colorectal cancer respectively[12,13]. In contrast, *Ruminococcus* is abundant in lung cancer with poor response to anti-PD-1 therapy[14]. Through the impact of the gut bacterial metabolism, diet influences the efficacy of tumor immunotherapy. However, much remains unexplored about the underlying mechanisms by which the diet-microbiome axis influences cancer immunotherapy.

Collectively, past mechanistic studies primarily focused on single factors from the diet, often overlooking the interactions of multiple factors carried by plant-derived nanoparticles that could directly affect gut microbiota function. The key points from this study have shown that GELN can enhance the efficacy of anti-PD-L1 therapies in melanoma by targeting the unsaturated fatty acids metabolic pathways of human gut commensal microbiota. This innovative strategy opens new avenues for improving cancer treatment outcomes by combining natural compounds with advanced immunotherapies.

## Results

### Plant-derived nanoparticles (PNP) are preferentially taken up by a specific human gut bacterial family

Food is digested when passing through the intestinal tract and releases nano-size particles into the gut lumen. Like exosomes, which are stable in the gut[15], the results published indicate that plant-derived exosomes-like nanoparticles (ELN) show a high tolerance to the gastrointestinal acidic environment, enzymes, and bile extracts[4,16–18]. The cargo of the edible nanoparticles may exert biological activity in the local gut environment or distance tissues by transfer from the gut lumen into the systemic circulation[19,20]. In this study, we mainly focused on investigating the impact of edible plant-derived nanoparticles (PNP) on the biological function of the human gut microbial community. As a proof of concept and based on the literature reports that PNP can be isolated from ginger root, garlic, aloe, and lemon[4,21,22], we began by characterizing PNP selected from these four plants. The PNP was extracted from ginger root, garlic, aloe, and lemon using differential centrifugation (Supplementary Fig. 1A) and purified using sucrose gradient centrifugation[4,23,24]. Bacteria and cellular debris were removed with centrifugation at $1000 \times$, $2000 \times$, and $4000 \times g$. Annexin V-coated magnetic beads (Miltenyi Biotec) were used to further remove potentially contaminating cellular debris. To determine whether plant-derived juice contains diverse nanoparticles, the pellets obtained from the centrifugation at $10,000 \times g$ were collected for the analysis of nanoparticle characterization[25,26], and the supernatants were collected for further exosome-like nanoparticles (ELN) isolation. The ELN was isolated with centrifugation at a final speed of

$100,000 \times g$[25,27]. The characteristics of the PNP in the pellets from the centrifugation at $10,000 \times g$ and $100,000 \times g$ were investigated subsequently and referred to as nanoparticles/$10,000 \times g$ (Nano10) and exosome-like nanoparticles/$100,000 \times g$ (ELN), respectively for this study. Each type of PNP including ELNs and Nano10 in this study was purified and collected from a predominant band after the sucrose gradient centrifugation (Supplementary Fig. 1B). All purified PNP from the corresponding predominant sucrose band was identifiable as nanoparticles based on transmission electron microscopy (TEM) examination according to their cup-shaped morphology and size (Supplementary Fig. 1C). Laser-based dynamic light scattering analysis with the NanoSight suggested that both ELN and Nano10 are similar in size based on diameter (150 nm to 250 nm), even if they are derived from different plants used in this study (Supplementary Fig. 1D). Although the size of Nano10 (diameter) is slightly smaller than the size of ELN[25,26], the difference is not statistically significant ($p > 0.05$). To investigate the mechanism underlying the finding that Nano10 exhibited a higher gravity in centrifugation even though they are a similar size compared to the ELN, we further estimated the density relative refractive index (RI) of PNP at a concentration of $1 \times 10^{12}$/ml using a refractometer (Leica, MARK II PLUS). The results indicated that there was no significant difference found in terms of the RI of ELN or Nano10 among the different plants (Supplementary Fig. 1E). However, when comparing the RI of ELN with Nano10 in all four plants, the RI of Nano10 is significantly higher than the RI of ELN ($1.46 \pm 0.21$ vs $1.35 \pm 0.14$, mean $\pm$ SD, $p = 3.18E\text{-}76 < 0.05$) (Supplementary Fig. 1E). This explains why Nano10 can be pelleted at $10,000 \times g$ while the ELN cannot. Since protein contributes the most to the cellular dry mass[28], we estimated the protein density in PNP using the weight of the protein in PNP divided by the volume of PNP. Interestingly, the protein density of Nano10 is dramatically higher than the protein density of the ELN in all four plants (Supplementary Fig. 1F), which is consistent with the refraction results. Moreover, the analysis of zeta potential using the ZetaView suggested the surface of ELN carries a more negative charge than the surface of the Nano10 (Supplementary Fig. 1G, H). The yield of each type of PNP measured by light scattering analysis (Supplementary Fig. 1I) suggested that edible plants could serve as a source for large-scale production of ELN and Nano10, especially grapefruit and ginger which have provided a higher yield of ELN and Nano10. Lemon had the lowest yield of PNP. Collectively, these data suggested that edible plants contain at least two types of PNP (Nano10, ELN) with different characteristics.

Many lines of evidence supporting the role of diet in regulating the composition and functions of the gut microbiota have been obtained from the murine gut microbe[4,20]. However, the composition of mouse gut microbiota is different from that of human gut microbiota[29,30]. To determine whether human gut bacteria take up PNP, fecal samples were collected from 15 healthy subjects (25–52 years old) with informed consent. Bacteria were subsequently isolated and pooled together to avoid the potential variation of bacterial species. An aliquot of human fecal bacteria (hFB) was colonized into germ-free (GF) C57BL/6 J mice (hFB mice) (Fig. 1A)[31–33]. The composition of fecal bacteria from 15 healthy subjects, small intestine (SI), and large intestine (LI) of hFB mice was determined using 16S ribosomal RNA (rRNA) sequencing (Supplementary Fig. 2A and Supplementary Tables 1, 2). The principal coordinate analysis (PCoA) on UniFrac distances of 16S sequencing profiles indicated that the composition of human fecal bacteria is ordinated closer to those translocated in mouse large intestine (LI) than those from mouse small intestine (SI) (Supplementary Fig. 2B).

To track PNP in vivo, the PNP isolated from edible plants (Supplementary Fig. 1) were labeled with fluorescence dye PKH26. The total PNP and PKH26+PNP were estimated using a NanoSight (NS300) with the filter wheel at clear and 565 nm (red fluorescence), respectively. The relative PKH26 labeling efficiency was calculated by dividing the

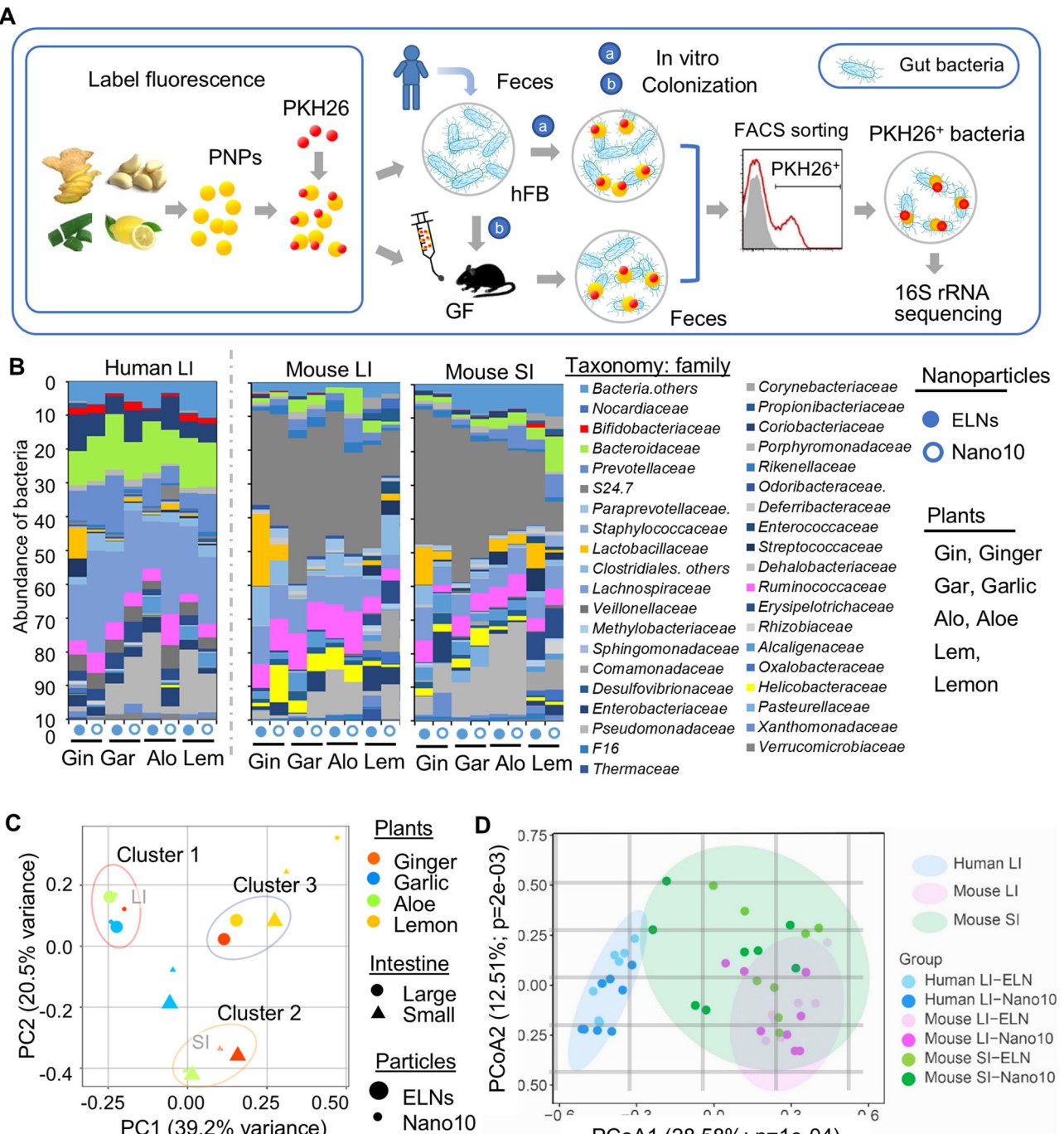

**Fig. 1 | Plant-derived nanoparticle (PNP) uptake by human gut bacteria.**
**A** Schematic representation of the processing of PNP, labeling with PKH26, and administration to germ-free (GF) C57BL/6 J mice colonized with human fecal bacteria (hFB) as well as PNP/PKH26+ bacteria analysis (n = 5 mice per group). Nano10 and exosome-like nanoparticles (ELN) collected from centrifugation at 10,000 × g and 100,000 × g, respectively. **B** The fecal bacteria from healthy subjects (n = 15) pooled together and incubated with PKH26-labeled PNP in an anaerobic chamber at 37 °C for 2 h (indicated as Human LI). An aliquot of human fecal bacteria colonized GF C57BL/6 J mice (hFB). PKH26-labeled PNP administered to mice at 0.5 g/kg (body weight, n = 6) for 2 h. Fecal bacteria collected from large intestine (indicated

as Mouse LI) and small intestine (indicated as Mouse SI). The PNP-PKH26+ bacteria from feces sorted by FACS, and DNA extracted for next-generation sequencing (NGS) of the 16S rRNA gene. The bar graph shows the percentage of taxonomy OTU in all sequence reads at the level of family. **C** Principal component analysis (PCA) of different sources and different types of PNP. The similar composition of bacteria is circled as a cluster. **D** Individual PNP-PKH26+ gut microbiota composition from human LI, hFB mice LI, and SI are shown on a principal coordinates analysis (PCoA) test according to their Bray-Curtis dissimilarity at the family taxonomic level. Source data are provided as a Source Data file.

PKH26+ PNP by the total PNP (Supplementary Fig. 3A left and middle). The result indicated that ginger and garlic-derived ELN and Nano10 have more than 50% labeling efficiency, but ELN or Nano10 from aloe and lemon have less than 50% labeling efficiency (Supplementary Fig. 3A left and middle).

Next, we determined whether PNP was taken up by gut bacteria. The uptake of nanoparticles by bacteria can occur quite rapidly, often within minutes to hours, depending on several factors, such as the type of bacteria, size, and materials of the nanoparticles[4,34]. Our data show that the PKH26 labeled PNP can be taken up by bacteria from human

feces after co-incubation at 37 °C for 30 min (Supplementary Fig. 3A **right**). To further confirm whether the taking up of the PNP takes place in vivo, the PKH26 labeled PNP were gavage administered to hFB mice. Since food can reach the large intestine as early as 1 h after being administered via oral gavage[35] and PNP can be taken up by human gut bacteria within 30 min, we assumed that two hours after gavage is sufficient for taking up PNP by bacteria. Therefore, the bacteria were isolated from the feces of mice that had been gavage-given for 2 h. PKH26[+] bacteria in the intestine (Supplementary Fig. 3B) were detected by confocal microscopy and quantified by FACS analysis (Supplementary Fig. 3C, D). To further identify the mechanism by which PNP is taken up by bacteria, ginger-derived ELN (GELN) was randomly chosen and labeled with PKH26 as an example and incubated with bacteria from human feces. The GELN/PKH26[+] signal in the bacteria was eliminated by the phagocytosis inhibitor cytochalasin D (Supplementary Fig. 3E), suggesting that GELN uptake by bacteria occurs via phagocytosis and this result excluded the possibility that the PKH26 fluorescent signal was generated from free form ELN/PKH26[+]. To estimate the stability of PNP in the stomach while passing through via oral gavage, the GELN and ginger Nano10 were exposed to gastric fluid collected from mice and incubated at 37 °C for three hours. The size and concentration of GELN and Nano10 were not altered, which indicates that the PNP is stable in the gastric fluid environment (Supplementary Fig. 4A). The polyacrylamide gel electrophoresis (PAGE) analysis further suggested that GELN cargo protein and RNAs especially small RNAs were not degraded under gastric fluid condition (Supplementary Fig. 4B, C). Collectively, the PNP from the four different types of plants can be taken up by human gut bacteria with a diversity of efficiency.

Next, to determine whether the PNP is preferentially taken up by a specific family of gut bacteria, the PNP labeled with PKH26[+] were administered to hFB mice via oral gavage, and the PNP/PKH26[+]bacteria were then sorted using fluorescence-activated cell sorting (FACS) Fig. 1A). To exclude potential bias in vitro and provide a mimic of gut environmental conditions, PKH26[+]PNP was administered to hFB mice by way of oral gavage and the fecal bacteria were isolated from the SI and LI two hours after gavage. The PKH26[+]PNP bacteria were then sorted by FACS (Supplementary Fig. 3C). Subsequently, the composition of PNP/PKH26[+] bacteria from in vitro and in vivo experiments was analyzed by next-generation sequencing of 16S rRNA (Fig. 1A, B). The sequencing data were deposited in the National Center for Biotechnology Information (NCBI) Gene Expression Omnibus (GEO) database with accession number GSE229897. A range of 102 k–185 k clean tags were obtained per sample for bacterial 16S rRNA genes through sequence optimization and quality filtering. At 0.03 similarity cutoff (97%), a subsample of 36 k 16S rRNA clean tags gave 615 operational taxonomic units (OTU). Using the Basic Local Alignment Search Tool (BLAST) search of the 16S sequences against the NCBI database, the OTUs were classified into 37 phyla, 65 classes, 115 orders, 192 families, 352 genus and other unidentified taxa at each taxonomic level. Comparing the composition of PNPs/PKH26[+] bacteria in vitro with the in vivo, our data indicated that the dominant PNPs/PKH26[+] bacteria families were *Lachnospiraceae* (21.48% ± 5.38%), *Bacteroidaceae* (13.01% ± 2.40%), *Coriobacteriaceae* (6.99% ± 2.67%) and *Ruminococcaceae* (4.92% ± 2.01%) in vitro. The in vivo results were S24.7 (32.43% ± 11.33%, LI; 21.87% ± 14.12%, SI), *Lachnospiraceae* (8.25% ± 4.62%, LI; 6.11% ± 4.29%, SI), and *Ruminococcaceae* (5.07% ± 2.47%, LI; 4.70% ± 2.97%, SI) no matter whether from the SI or LI (Fig. 1B, Supplementary Fig. 5A–Cand Supplementary Tables 3–5). Interestingly, for *Ruminouptakeae* up take of PNPs, there was no difference between in vitro and in vivo. Collectively, these results suggest that PNP can be taken up by gut bacteria, and except for Ruminococcaceae the environment (in vitro versus in vivo) has an impact on the selectivity of the uptake of PNP. Moreover, we found that the uptake efficiency was dependent on the type of PNP (ELN versus Nano10) and types of plants used for the isolation of PNP (Fig. 1B, Supplementary Fig. 5A–C and Supplementary Tables 3–5).

We then investigated whether PNP is taken up by closely related families of gut bacteria. A principal component analysis (PCA) was performed to explore the variation in PNP recipient bacterial composition among a selection of PNP and different locations in the intestine (Fig. 1C). The PCA biplot captured most of the variance within the bacterial composition along the first two principal components. Principal component 1 (PC1) accounts for 39.2% of the variance, while Principal component 2 (PC2) explains an additional 20.5% of the variance, cumulatively covering over 59% of the total variability in the data (Fig. 1C). Multivariate statistical PCA analysis revealed that there are 3 clusters based on the components i.e., bacterial composition close to each other (Fig. 1C). Cluster 1 includes ELN and Nano10 of garlic, ginger Nano10 and aloe ELN that are preferentially taken up by related similar bacteria in the LI. Cluster 2 includes aloe- and ginger-derived PNP that share similar recipient bacteria in the SI. In Cluster 3, similar gut bacteria preferentially take up ginger- and lemon-derived ELN in the LI, as well as lemon-derived ELN in the SI. To test the influence of bacterial environment on PNP uptake, we then applied principal coordinates analysis (PCoA) analysis on UniFrac distances of OTUs at the family level grouped as in vitro (human LI) and in vivo (mouse LI and SI). Our analysis revealed that the distance between samples was primarily determined by PCoA1 and PCoA2 (Fig. 1D). The statistical analysis suggested a significant difference in the principal component 1 (28.58%, $p = 0.0001$) and principal component 2 (12.51%, $p = 0.002$) among these three conditions. Taken together, our data suggested that diet-derived PNP can be selectively taken up by human gut bacteria, and multiple factors influence the bacteria uptake including the type of PNP (Nano10, ELN), type of plants used for isolation of PNP, and the intestinal microenvironment.

## PNP lipids and amino acids serve as "Eat me"/"Don't eat me" signals for human gut bacteria

Diet plays a crucial role in modulating gut microbiota homeostasis with numerous functional nanoparticles[36,37]. Emerging studies[38,39] and our data (Fig. 1) suggest that edible PNP are selectively taken up through phagocytic pathways by gut bacteria and PNP-derived factors such as lipids and amino acids (AA) present a positive or negative impact on gut bacteria taking up nanoparticles to modulate gut microbiota homeostasis. Further, the molecular mechanism underlying how PNP can be selectively taken up by gut bacteria via "eat me"/ "don't eat me" signals were elucidated. We previously demonstrated that *Lactobacillaceae* was preferentially taken up by GELN in the murine gut and was attributed to highly enriched phosphatidic acid (PA) in the GELN[4]. However, whether PNP lipids play a causative role in selective uptake by gut bacteria in general is not known. To determine the PNP lipid role, lipids extracted from the PNP listed in Supplementary Table 6 and the lipid composition were identified by lipidomic analysis using triple quadrupole mass spectrometry (MS). The composition of ELN and Nano10 isolated from 4 different types of plants is listed (Fig. 2Aand Supplementary Table 6). The Kruskal–Wallis H test and violin analysis revealed a significantly different composition of lipids between ELN and Nano10 ($p = $1e-09), specifically, PA is enriched in ELN, and lyso-phosphatidylglycerol (lysoPG) and digalactosyldiacylglycerol (DGDG) are enriched in Nano10 regardless of which types of plants the PNP is derived (Fig. 2B). The PCA plot of the lipid profiles further revealed that the overall lipid composition of garlic PNP and lemon PNP is more similar since they are closer to each other on the PCA plot compared to the PNP from other types of plants (Fig. 2C). In contrast, the composition of ELN-derived lipids in ginger, aloe, and garlic are distinguished from their Nano10's, which indicates more variation (Fig. 2C).

Next, we determined whether PNP-derived factors can serve as "eat me"/"don't eat me" signals for PNP to be selectively taken up by

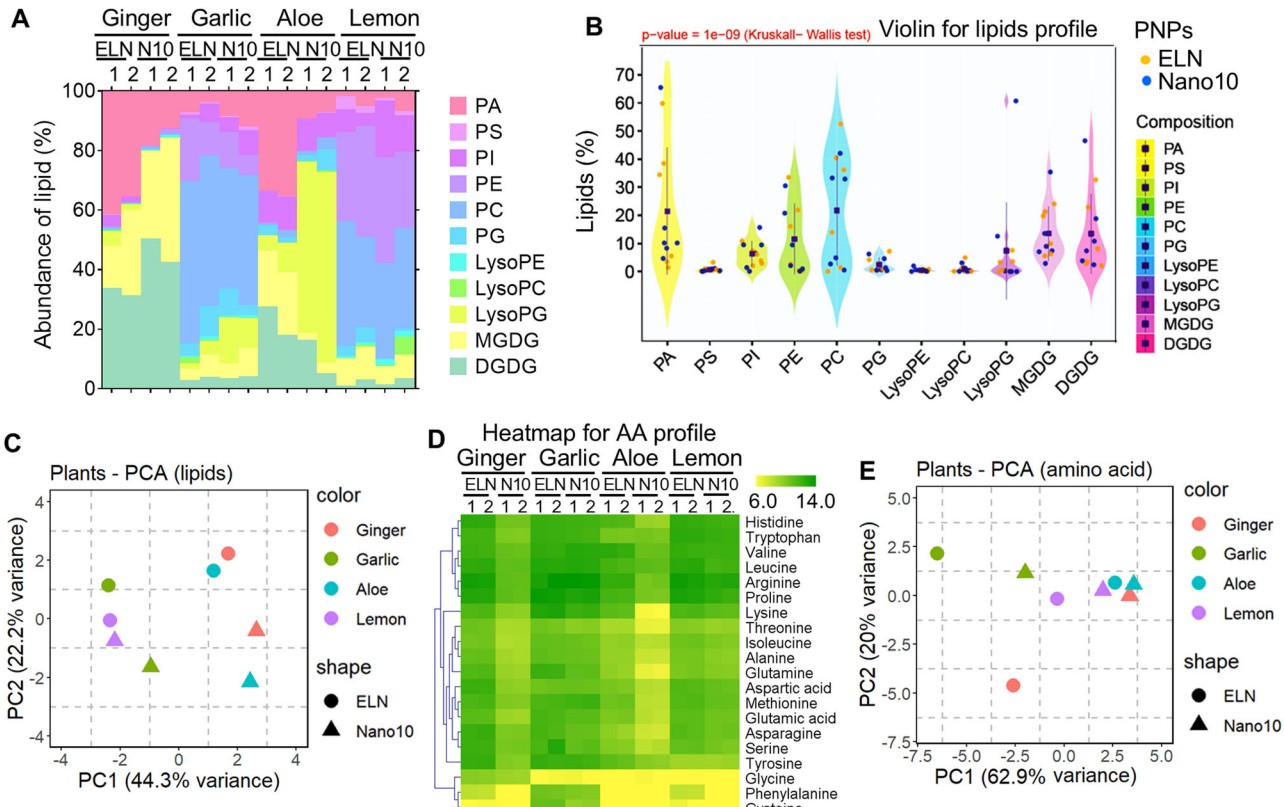

**Fig. 2 | Composition analysis of PNP lipids and amino acids. A** Lipid composition of PNP identified with the liquid chromatography-mass spectrometry (LC-MS) (*n* = 2, each group was pooled from three independent samples). N10, Nano10. **B** Violin plots showing the composition of lipids in ELN (orange) and Nano10 (blue). The dots represent the mean proportion of each lipid in ELN and Nano10. Vertical lines indicate the interquartile range. Statistical significance between ELN versus

Nano10 is indicated in the plot at *p* < 0.01 using the Kruskal–Wallis test in R. **C** Principal component analysis (PCA) of lipid composition in PNP using R. **D** Heatmap indicating amino acid (AA) level on the PNP membrane identified by LC-MS. N10, Nano10. **E** Principal component analysis (PCA) of AA levels in PNP using R. Source data are provided as a Source Data file.

gut bacteria. Our published data indicate that PA-enriched ginger ELN contributes to *Lactobacillaceae* uptake[4]. However, *Lactobacillaceae* took up very few of the aloe-ELN (AELN) (Fig. 1B) although AELN has a high level of PA (Fig. 2A), suggesting that more than one factor in PNP may serve as a signal mediating gut bacterial uptake. Given that nanoparticle membranes are composed of multiple functional factors, including lipids, nucleic acid, and protein/AAs[40], and that AAs were found to be selectively taken up by bacteria[41], we hypothesize that besides lipids, AAs on the PNP membrane also as "eat me"/"don't eat me" signals. To address our hypothesis, we first assessed the composition of all twenty AAs on the PNP membrane using MS analysis. We found that the PNP membrane has a higher level of histidine, tryptophan, valine, leucine, arginine, and proline than other AAs (Fig. 2D and Supplementary Table 7). ELN contained higher levels of AAs on their membrane in general compared to Nano10. Specifically, GELN and ginger-nano10 contain a relatively high level of glycine whereas garlic PNP contains a relatively high level of phenylalanine and cysteine (Fig. 2D). A principal component analysis (PCA) was performed to explore the variation in AAs composition among PNPs. The PCA biplot (Fig. 2E) captures more than 80% of the variance within the AAs profiles along the first two principal components. Principal Component 1 (PC1) accounts for 62.9% of the variance, while Principal Component 2 (PC2) explains an additional 20% of the variance. The Nano10 from lemon, aloe, and ginger has a common distribution of AAs because they are closer to each other on the PCA plot. Only aloe has a similar AAs composition on its ELN and Nano10 (Fig. 2E).

To explore whether PNP lipids and AAs can serve as "eat me"/ "don't eat me" signals for enhancing/preventing uptake by gut

bacteria, a correlation analysis between the relative composition of the lipids and AAs from all PNP and their enhancing/preventing of uptake by bacteria was assessed using Spearman's correlation coefficient test. Clustered heatmaps with the K-Means Clustering method[42] were used to show the patterns of these correlations with the correlation coefficients in distinct gut locations (Fig. 3A and Supplementary Table 8) and integrated multiple locations. Red and blue colors indicated positive and negative correlation coefficients, respectively, between each gut bacteria taxonomic family and the content of PNP lipids/AAs. The volcano plots (Fig. 3B, C) were generated to display the statistical significance of the association against the correlation coefficient. These plots serve as a visual representation of the correlation between PNP lipids (Fig. 3B)/PNP AAs (Fig. 3C) and bacterial abundance. The volcano plots were stratified to highlight statistically significant correlations, using *p*-value thresholds of 0.05. The top 10 correlations based on p-value ranking were listed aside from the plots (Fig. 3B, C), e.g., lipid monogalactosyl diacylglycerol (MGDG) enriched PNP serve as an "eat me" signal for *Lactobacillaceae* preferential uptake and phosphatidylinositol (PI) enriched PNP serves as "do not eat me" signal to prevent *Lactobacillaceae* uptake. *Lactobacillaceae* also preferentially take up glycine-enriched PNP, and valine inhibits PNP uptake by *Ruminococcaceae* and *Lachnospirasceae*. All of these correlation analyses agree with data (Fig. 1B) from the in vitro human LI, in vivo mouse LI, and mouse SI. The independent correlation analysis for each condition indicated the influence of the microenvironment (SI versus LI) on the bacteria uptake of PNP (Supplementary Fig. 6A–C).

To experimentally verify our analysis of the correlation, suggesting that lipids of PNP can enhance/inhibit uptake by gut bacteria, we

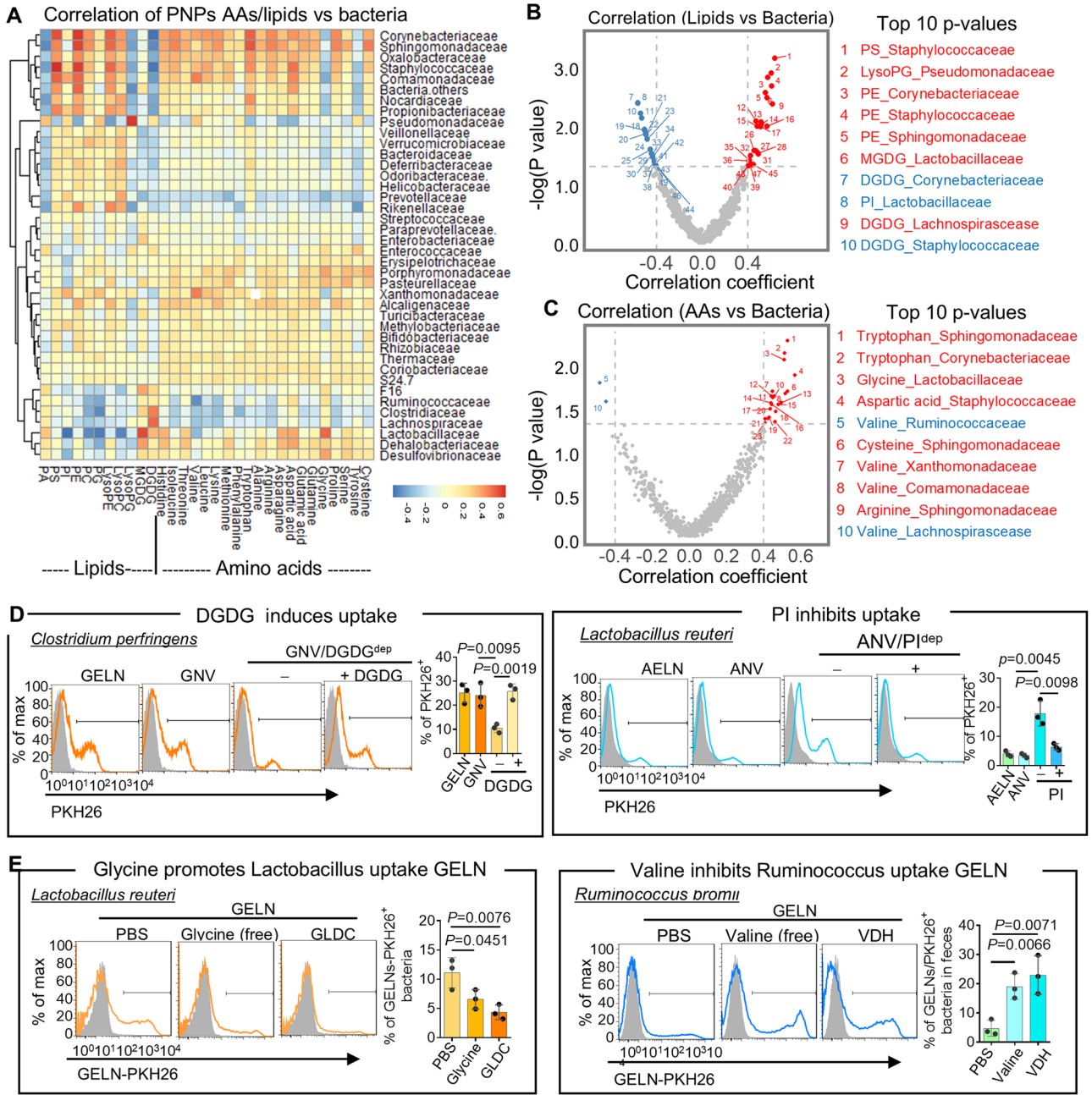

**Fig. 3 | Correlation analysis of gut bacteria preferentially taken up and the composition of PNP lipids and amino acids. A** Heatmap indicating Spearman's correlation coefficients between the lipid compositions or AA levels in PNP and the PNP recipient bacteria integrated from Human LI, Mouse LI, and Mouse SI in Fig. 1B. **B, C** Volcano plots of Spearman correlation coefficients ($x$-axis) and significance ($y$-axis) showing a correlation between the lipids (**B**) or AA levels (**C**) in PNP and the composition of PNP recipient bacteria. Significant associations ($P < 0.05$) are highlighted. List of top 10 $p$-values in correlation test aside from the plots. **D** DGDG and PI depleted from ginger and aloe-derived nanovesicles (GNV & ANV) respectively. FACS analysis of *Clostridium perfringens* (*C. perf*) incubated with PKH26-labeled GELN, GNV with DGDG depletion (DGDG$^{dep}$) and supplemented with DGDG

(left panel), and *Lactobacillus reuteri* (*L. reuteri*) incubated with PKH26-labeled AELN, ANV with PI depletion (PI$^{dep}$) and supplemented with PI (right panel). Quantification of PKH26$^+$ bacteria (bar graphs). ($P = 0.0095$, $P = 0.0019$, $P = 0.0045$, $P = 0.0098$, two-way Chi-Square test, $n = 5$). **E** *L. reuteri* and *Ruminococcus bromii* pretreated with glycine (1 mg/ml) and valine (1 mg/ml) at 37 °C for 1 h, respectively, following incubation with glycine dehydrogenase (GLDC) (left panel) or valine dehydrogenase (VDH) (right panel) treated GELN/PKH26. FACS analysis of PKH26$^+$ bacteria. Quantification of PKH26$^+$ bacteria (bar graphs). ($P = 0.0451$, $P = 0.0076$, $P = 0.0066$, $P = 0.0071$, two-way Chi-Square test, $n = 5$). Data are representative of three independent experiments as the mean ± standard deviation (SD, error bars). Source data are provided as a Source Data file.

conducted the uptake tests with the specific species of bacteria. Due to the commercial availability of specific species of bacteria, we used *Clostridium perfringens* (*C. perf*), a species in the Lachnospirascease family, and *Lactobacillus reuteri* (*L. reuteri*), a species in the Lactobacillaceae family to test our lipids correlation results. We randomly picked up DGDG ("eat me") and PI ("do not eat me") as the *Lachnospirascease* family preferentially takes up DGDG-enriched GELN, and PI-

enriched AELN inhibits *Lactobacillaceae* uptake of AELN (Fig. 3A–C). To test whether PNP lipids mediate bacteria uptake, the DGDG and PI were depleted from the total lipids of GELN and AELN, respectively[4], and the remainder of the lipids were reassembled into a nano-sized nanovesicle (NV) by sonication with the protocol we published[4] and referred to them as ginger NV(GNV) and aloe NV (ANV), respectively. The FACS analysis indicated that GELN and GNV, AELN and ANV have

similar uptake efficiencies by recipient bacteria (Fig. 3D). Moreover, the depletion of DGDG from the GNV lipids significantly reduced *C. perf* uptake of GNV whereas the addition of DGDG rescued the uptake of GNV (Fig. 3D, **left panel**). In contrast, the depletion of PI promotes the uptake of ANV by *L. reuteri* and the addition of PI restored the inhibitory effect on the uptake of ANV (Fig. 3D, **right panel**).

To experimentally verify whether AAs of PNP can enhance/inhibit uptake by gut bacteria, we conducted the uptake tests with the specific species of bacteria. Due to the commercial availability of specific species of bacteria, *Lactobacillus reuteri* (*L. reuteri*), a species in the Lactobacillaceae family, and *Ruminococcus bromii* (*R. bromii*), a species of the *Ruminococcaceae* family were used to test our AAs correlation results. We randomly picked up glycine ("eat me") and valine ("do not eat me") as GELN glycine promotes *Lactobacillaceae* taking up GELN and valine inhibits Ruminococcaceae family uptake of GELN (Fig. 3A–C). To confirm our AAs correlation results, *L. reuteri,* and *R. bromii* were preincubated with glycine and valine, respectively. The FACS analysis demonstrated that preincubation of the free form of glycine inhibits *L. reuteri* uptake of GELN, whereas preincubation of the free form of valine promotes *R. bromii* uptake of GELN (Fig. 3E, **left panel**). These conclusions are also supported by the facts that the depletion of glycine from GELN with glycine dehydrogenase (GLDC) reduced *L. reuteri* uptake of GELN whereas depletion of valine from GELN with valine dehydrogenase (VDH) induced *R. bromii* taking up GELN (Fig. 3E, **right panel**). These results are also consistent with the analysis of correlation suggested that PNP-derived multiple factors, including lipids and AA, can serve as an "eat me"/"don't eat me" signal for enhancing/inhibiting taken up by human gut bacteria.

Given different types of edible plants contain a variety of PNP, and each type of PNP has its unique composition of lipids and AAs, our data suggest that specific lipids and AAs profiles on PNP can regulate the homeostasis of gut microbiota via selectively enhancing/inhibiting PNP up taken by gut bacteria.

## PNP modulates the cellular metabolic pathways by altering metabolites of human gut bacteria

The role of microbiota in health and diseases has been highlighted by numerous studies, and gut microbiota-derived metabolites have a central role in the physiology and physiopathology of metabolic disorders[43]. Next, we proposed that upon PNP uptake by specific species of gut bacteria, the composition of the metabolites released from PNP recipient gut bacteria may be altered. To identify the human gut bacterial-derived metabolites that are altered by PNP, we compared the composition of gut metabolites released from GF mice and hFB mice treated for three days with different types of PNP (0.5 g, PNPs weight/kg of mouse body weight per day) via oral gavage. The fecal samples from the LI and SI were collected and suspended in PBS. The supernatant was isolated after centrifugation for analysis of metabolites. 2D- liquid chromatography-tandem MS (LC-MS/MS) analysis indicated that PNP modulated the composition of gut metabolites in GF mice (Supplementary Fig. 7A and Supplementary Data 1) and hFB mice (Supplementary Fig. 7A and Supplementary Data 2). To focus on the bacterial metabolites and exclude the impact of host metabolites, the LC-MS/MS data from hFB mice normalized by the results of GF mice represented the alteration of bacterial metabolites. Heat mapping analysis showed the top up- and down-regulated gut bacterial metabolites modulated by PNP. The data suggested that the effects of PNP derived from ginger (Fig. 4A), garlic (Fig. 4B), aloe (Fig. 4C), and lemon (Fig. 4D) on the levels of bacterial metabolites are dependent on the type of PNP and the gut microenvironment. For example, the ELN and Nano10 from the same type of plants have different effects. Aminobutyric acid (GABA) was highly enriched in ELN-treated gut bacterial metabolites but not in Nano10-treated bacteria. As an inhibitory neurotransmitter, GABA has antimicrobial, antiseizure, and antioxidant properties[44]. This data suggests that ELN could potentially be used for

the induction of GABA but not by Nano10. The gut bacteria-derived unsaturated fatty acids (USF) including eicosapentaenoic acid (EPA) and docosahexaenoic acid (DHA) were significantly induced by GELN in the SI but not in the LI and not by Nano10 or other plant-derived PNP (Fig. 4A). This result suggests that the gut microenvironment influences PNP mediated regulation of production of metabolites released from human gut microbiota.

We next sought to access human metabolism-based pathways that are potentially regulated by human gut bacteria-derived metabolites modulated by PNP. To determine which metabolic pathways in the mammalian cell have been altered due to ELN and Nano10 treatment, the KEGG metabolomics database was used. Using the most up-to-date KEGG metabolomics database, 166 differential metabolites from gut bacteria were found to be affected by ELN and Nano10 and have been linked to human metabolism-based pathways. The metabolic pathways were visualized in scatter plots (Fig. 5A). The relative impact of the metabolic pathway was estimated by the fold-change of all metabolites in the pathway and is indicated in Fig. 5A. More specifically, the results suggested that the metabolites from the LI have significant effects on the alanine, aspartate, glutamate (AAG) pathways due to ELN treatment, whereas the tricarboxylic acid (TCA) cycle pathway was affected by Nano10 regardless of the type of plant. Interestingly, in the SI, both the AAG and TCA pathways are highly impacted by ginger and garlic-derived PNP. However, aloe ELN and lemon ELN have no effect on the TCA and AAG pathways, respectively, whereas ginger ELN significantly up-regulates the USF pathway in the SI. A heat map (Fig. 5B) indicates that the activities of the D-glutamine and D-glutamate (DGG), arginine and proline (ARP), and taurine and hypotaurine (THT) pathways are enhanced by all four ELNs, whereas the activity of these pathways was eliminated by all four of the Nano10. In contrast, glutathione (GTA) activity was decreased with all four ELNs and increased by all four of the Nano10. The change could not be due to variation of amounts of nanoparticles used for the treatment of hFB mice as there was no difference in terms of ANS, ASC, and GPL activities regardless of which types of nanoparticles were used for the treatment of the mice. Also, we noticed that the biotin metabolism (BIO) pathway was only altered in the LI of mice and multiple pathways including BAL, AAT, MTB, USF, HIT, and PTG were altered in the SI after PNP treatment (Fig. 5B). Collectively, these data suggest that ELN and Nano10 likely regulate the homeostasis of human metabolism-based pathways such as DGG and THT pathways via the metabolites released from gut bacteria targeted by ELN and Nano10.

To verify the results from high throughput 2D-LC-MS/MS analysis, randomly selected metabolites from the gut of GF and hFB mice were further evaluated by individual HPLC. Consistently, the ELN from ginger, garlic, aloe, and lemon dramatically induced glutamine and glutamic acid that involves the pathways of AAG, DGG, and AAT (Supplementary Fig. 7B). The levels of hypoxanthine and 5-GMP involved in the pathway of purine (PUR) were decreased by the ELN from ginger, garlic and aloe, but induced by the Nano10 from aloe and lemon (Supplementary Fig. 7C). Collectively, the analysis of the metabolome suggests that even with the diversity of PNP, they have a common impact on some bacteria metabolites and their regulatory pathways. In addition, some PNP have a unique impact on bacterial metabolism. These findings provide the rationale for choosing specific ELN or Nano10 as therapeutic delivery agents to treat diseases with specific pathway dysregulation.

## GELN RNAs modulate gut bacterial metabolite activity by contributing to the inhibition of melanoma growth and metastasis in a mouse model

Next, we determined whether shaping gut bacterial metabolites by PNP benefits for the treatment of dysbiosis-related disease. Since immunotherapy has only benefited a subset of patients due to the heterogeneity of the patients' microbiome[11,45], the immune checkpoint

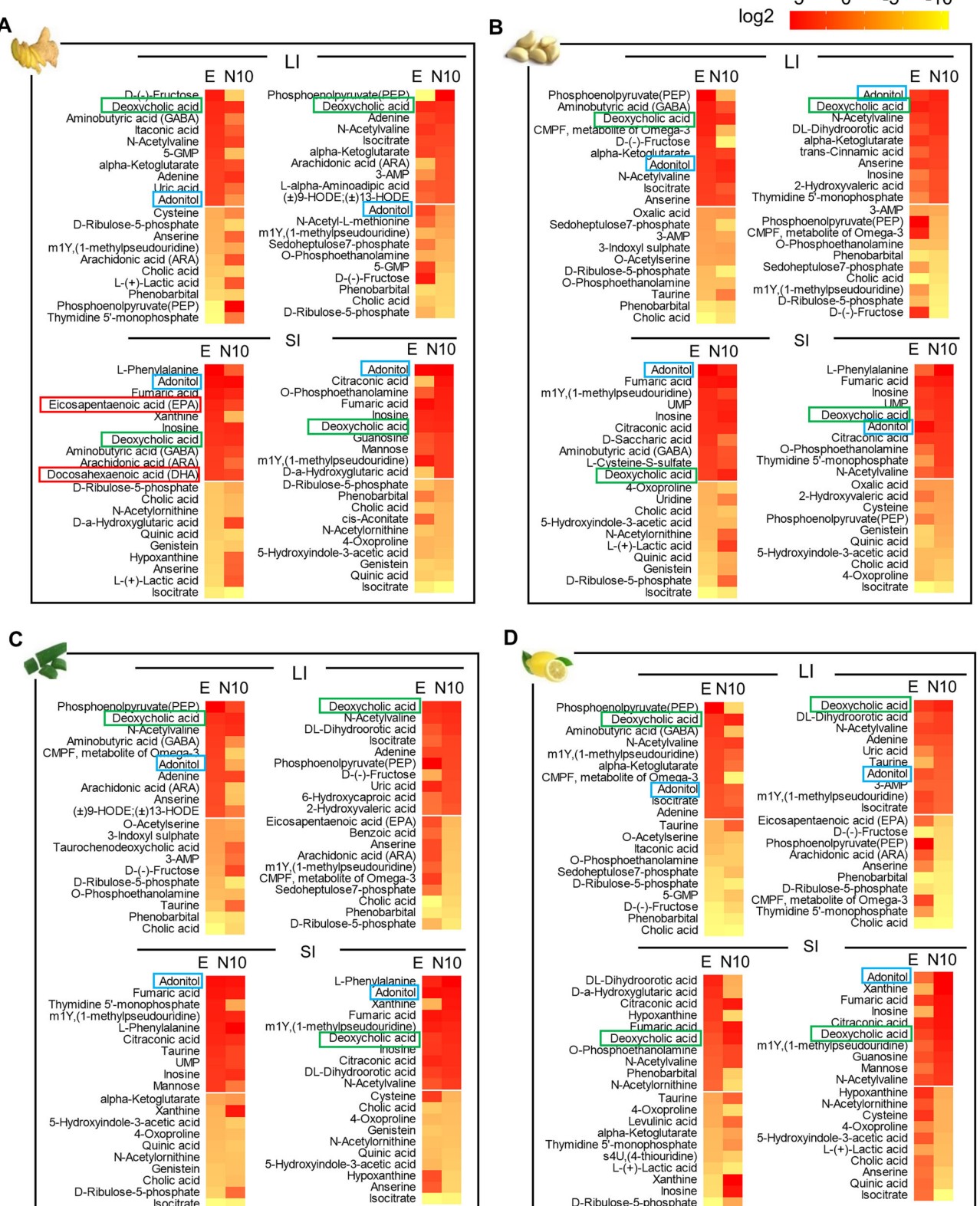

**Fig. 4 | The influence of PNP on the gut bacterial metabolites using metabolomics analysis. A**–**D** GF mice and hFB mice treated with PNP (0.5 g/kg, body weight) every other day for two months. Gut feces from GF and hFB mice suspended in PBS (1.0 g/ml). After centrifuging at 10,000 × *g*, the supernatant was used for metabolomics analysis using 2D-LC-MS/MS. Heatmap indicating top 10 bacterial metabolites up- or down-regulated by ELN (E) and Nano10 (N10) in hFB mice normalized to GF. Source data are provided as a Source Data file.

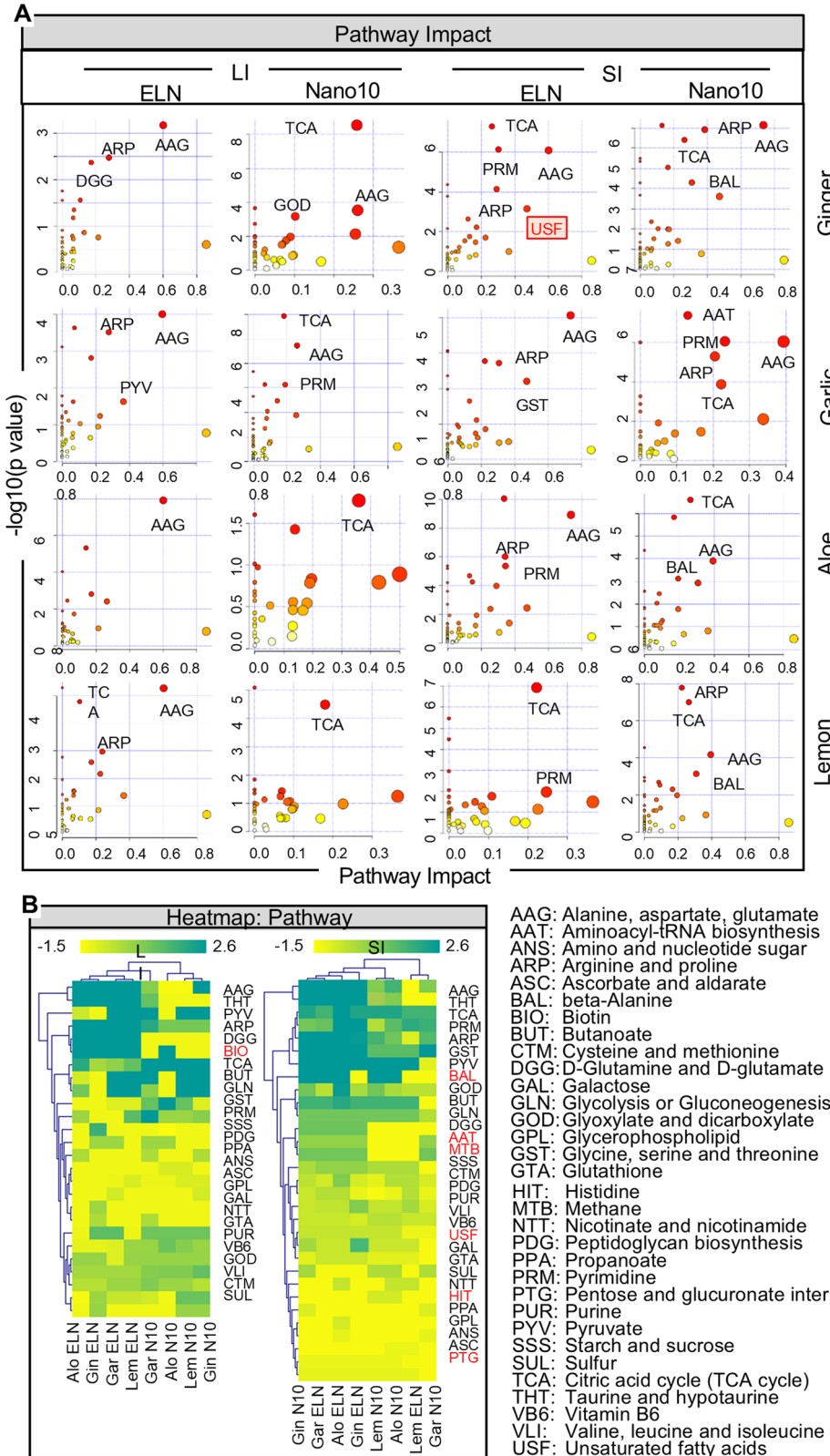

**Fig. 5 | The PNP-mediated influence of bacterial metabolites on host metabolic pathways. A** The relative impact on the microbiome metabolic pathway by PNP analyzed with MetaboAnalyst. **B** Heatmap of the metabolomics pathway in human gut bacteria colonized GF mice treated with PNP using gplots (R package). Source data are provided as a Source Data file.

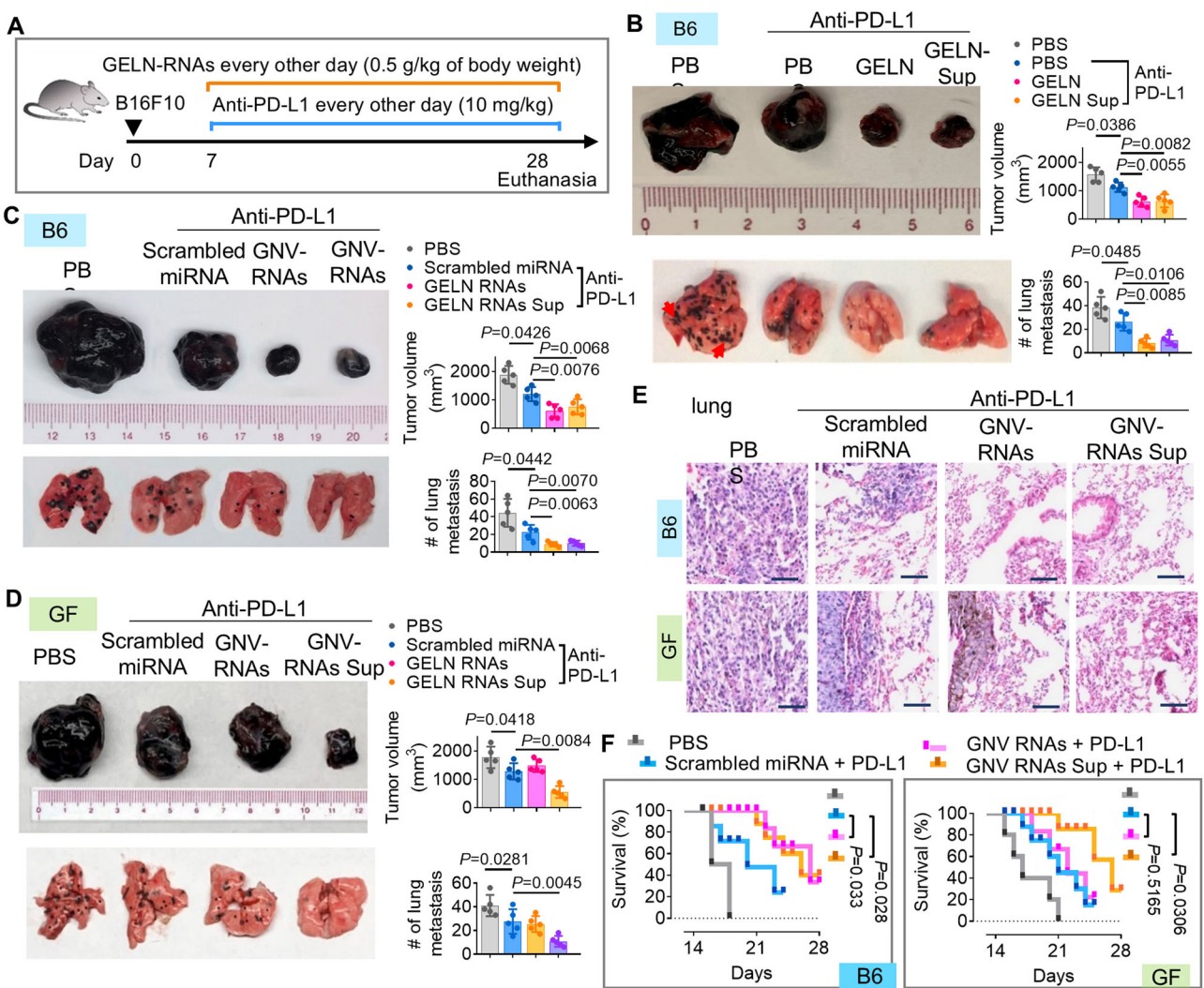

**Fig. 6 | Ginger-ELN (GELN) improves PD-L1 antibody effect against melanoma mediated by gut metabolites. A** Schematic representation of the treatment schedule for B16F10 melanoma cell (1 × 10⁵, *n* = 5) inoculation of hFB mice with anti-PD-L1 antibody (Bioxcell BE0101, 15 mg/ml) via intraperitoneal (IP) injection as well as oral administration with GELN or gut fecal Sup containing metabolites (100 µl from 100 mg feces) from GELN treated mice. **B** Representative B16F10 melanoma primary tumor (top left) and lung (bottom left, metastatic nodules indicated by red arrows) from tumor-bearing hFB mice (*n* = 5) at 28 d post-injection of B16F10 cells along with anti-PD-L1 antibody w/o GELN or gut metabolites from GELN treated mice. Quantification of primary tumor volume and metastatic nodule number (>1 µm) (right panel). (*P* = 0.0386, *P* = 0.0055, *P* = 0.0082; *P* = 0.0485, *P* = 0.0085, *P* = 0.0106; two-way ANOVA test, *n* = 5). **C** Representative examples of B16F10 melanoma primary tumor (top left) and lung (bottom left) from tumor-bearing hFB

mice (*n* = 5). The treatment of GELN replaced with GELN-RNAs (0.5 mg/kg, body weight) encapsulated with GELN-derived nanovesicle (GNV). (*P* = 0.0426, *P* = 0.0076, *P* = 0.0068; *P* = 0.0442, *P* = 0.0063, *P* = 0.0070; two-way ANOVA test, *n* = 5). **D** Representative examples of B16F10 melanoma primary tumor (top left) and lung (bottom left) from tumor bearing GF mice (*n* = 5). The treatment of each group was the same as in (**B**). (*P* = 0.0418, *P* = 0.0084; *P* = 0.0281, *P* = 0.0045; two-way ANOVA test, *n* = 5). **E** Representative hematoxylin and eosin (H&E)-stained sections of melanoma tumor tissue (200x magnification) from tumor-bearing mice from (**C**) and (**D**). Scale bars, 200 µm. **F** Survival rate of mice in hFB (**C**) and GF (**D**) mice. (*P* = 0.033, *P* = 0.028; *P* = 0.5165; *P* = 0.0306; two-way Chi-Square test, *n* = 5). Data are representative of three independent experiments as the mean ± standard deviation (SD, error bars). Source data are provided as a Source Data file.

inhibitor PD-L1 against melanoma growth and metastasis was used to further test our hypothesis that PNP could enhance anti-PD-L1 based immunotherapy mediated by PNP targeting of the gut microbiome. The hFB colonized male mice were given PNP for one month via oral gavage and the gut metabolites were then collected from the fecal supernatant (PNP-Sup). The B16F10 melanoma cells were subcutaneously injected into hFB mice. After 7 days of inoculation, anti-PD-L1 mAb (Bioxcell BE0101, 10 mg/kg, body weight) was administered to melanoma B16F10 bearing mice via intraperitoneal (IP) injection, along with PNP or PNP-Sup (Fig. 6A). We found that ginger-derived ELN (GELN) or GELN-Sup dramatically enhanced the anti-PD-L1 therapy against B16F10 melanoma growth (Fig. 6B, **top panel**) and lung metastasis (Fig. 6B, **bottom panel**). However, the same plant-derived

Nano10 (Ginger-Nano10), garlic-ELN, aloe-ELN, and lemon-ELN had no effect on the anti-PD-L1 therapy in melanoma growth in mice (Supplementary Fig. 8A).

Considering that ELN is enriched with small microRNA (miRNA) and that in our previous study of GELN-RNA tryptophan metabolism, the gut bacteria were influenced[4], we next sought to determine whether GELN-RNAs played a role in the melanoma responding to anti-PD-L1 treatment. To exclude the influence of other factors such as small molecules and proteins in GELN, GELN lipids were extracted and assembled into GNV. The GELN-derived RNA was isolated and encapsulated back into GNV (GNV-RNA) under ultrasonication[4,23]. Given that small RNAs dominate the ELN cargo[4], we used a scrambled miRNA as the control. Our results indicated that both GNV-RNA and gut

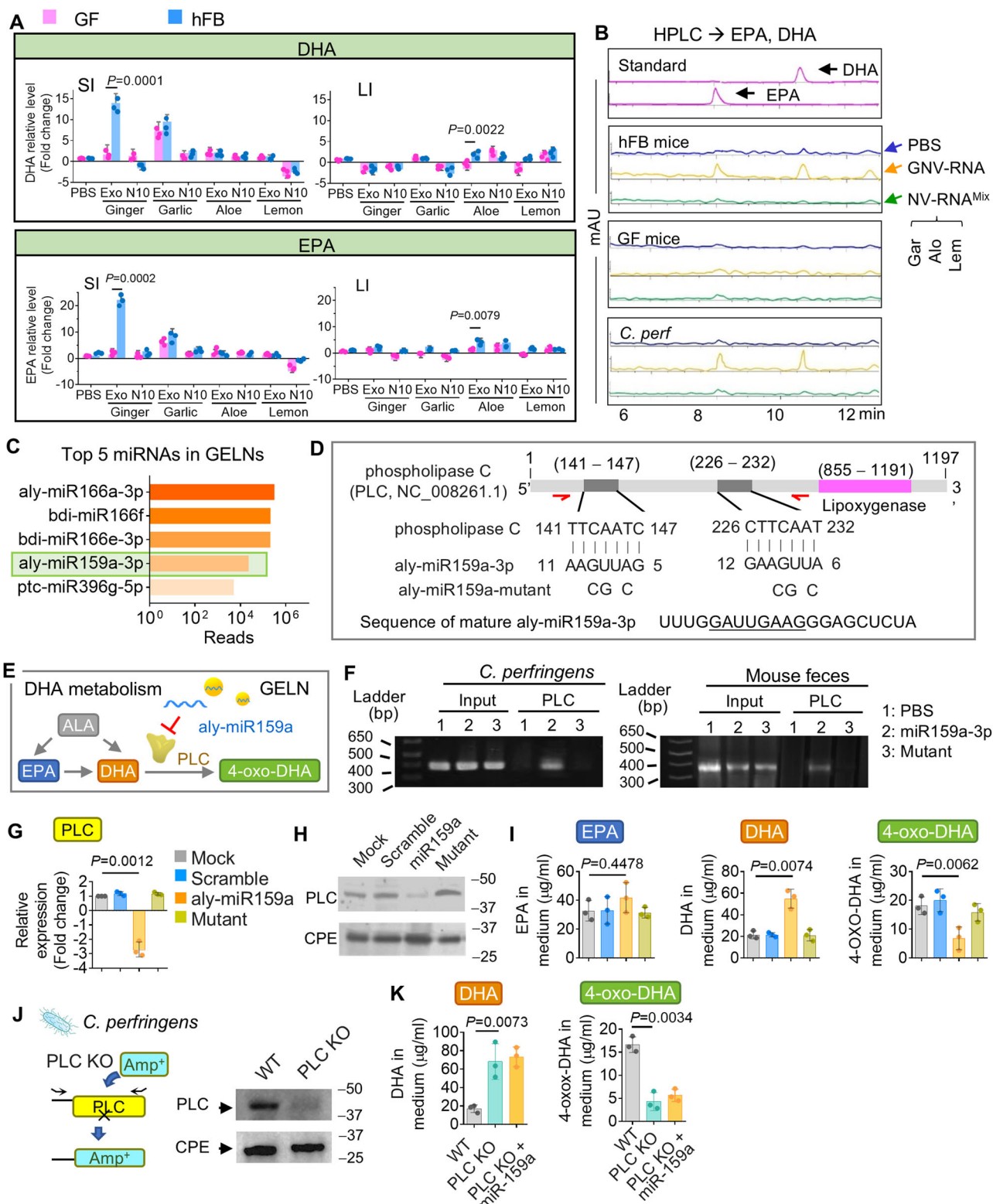

metabolites from GNV-RNA fed hFB mice (GNV-RNAs-sup) exhibited a similar effect on enhancing anti-PD-L1 mediated effects against B16F10 melanoma advancement (Fig. 6C) and on the induction of cleaved caspase 3, cleaved PARP (Supplementary Fig. 8B). To confirm that the metabolites contributed to anti-PD-L1 mediated therapy against tumor progression were associated with gut microbial metabolism, we assessed the response of B16 F10 melanoma bearing GF mice receiving anti-PD-L1 and GNV-RNA treatments. There was no evidence that GNV-RNAs enhanced anti-PD-L1 mediated therapy against B16F10

melanoma, but gut metabolites from GNV-RNAs fed specific-pathogen-free (SPF) hFB mice recovered the activity of GNV-RNAs mediated enhancing of anti-PD-L1 therapy against melanoma growth and lung metastasis (Fig. 6D). The comparative studies suggested that GNV-RNAs improve anti-PD-L1 therapy against melanoma lung metastasis (Fig. 6E) and prolong survival (Fig. 6F) in hFB mice but not GF mice. Without anti-PD-L1 antibody treatment, hFB mice treated with GNV-RNAs or GNV-RNAs or gut metabolites from mice treated with GNV-RNAs have no significant benefit in terms of melanoma tumor growth

**Fig. 7 | GELN-enriched with aly-miR159a-3p modulates metabolism of DHA in bacteria by targeting PLC. A** Relative level of docosahexaenoic acid (DHA) and eicosapentaenoic acid (EPA) in LI and SI of GF and hFB mice, respectively, using HPLC ($n = 3$). ($P = 0.0001, P = 0.0022; P = 0.0002, P = 0.0079$, two-way ANOVA test, $n = 3$). **B** HPLC analysis of DHA and EPA in the gut feces from GF and hFB mice as well as medium of *Clostridium perfringens (C. perf)* (*C. perf*, ATCC 13124) treated with GNV-RNAs (0.5 mg/kg, body weight; 10 mg/1 × 108 bacteria) and ELN-RNAs mixed from garlic, aloe, and lemon. **C** Top five abundance miRNAs in GELN from NGS miRNA sequencing. **D** Schematic diagram of the putative binding sites of aly-miR159a-3p in the phospholipase C (PLC) of *C. perf*. Red arrows indicate primers for ChIP. The pink bar indicates the lipoxygenase domain. The aly-miR159a-3p seed matches in the PLC gene mutated at the positions indicated. **E** Schematic representation of DHA metabolism derived from EPA and α-linolenic acid (ALA) and converted to 4-oxo-DHA by PLC. The hypothetical model of GELN-derived aly-miR159a-3p regulates the metabolism of DHA and EPA by targeting PLC. **F** Analysis of interaction between miR159a-3p and PLC gene using the ChIP assay. Biotin-labeled aly-miR159a-3p incubated with DNA from *C. perf* (left panel) and feces from the colonized hFB mouse (right panel). PLC gene binding to aly-miR159a-3p pulled down with streptavidin beads and amplified with specific PLC primers. **G** qPCR analysis of PLC expression in *C. perf* treated with scrambled miRNA, aly-miR159a-3p, and miR159a mutant. ($P = 0.0012$, two-way ANOVA test, $n = 3$). **H** Western blot analysis of PLC expression. *C. perf* enterotoxin (CPE) served as a loading control. The size (kDa) of protein molecular weight (MW) indicated. **I** HPLC analysis of EPA (left), DHA (middle), and 4-oxo-DHA (right) in *C. perf* treated with aly-miR159a as well as control miRNAs. ($P = 0.4478, P = 0.0074, P = 0.0062$, two-way ANOVA test, $n = 3$). **J** Schematic diagram of PLC knockout (KO) in *C. perf* and replaced with ampicillin-resistant (Amp$^+$) sequence using fusion PCR (left panel). Western blot analysis for wildtype (WT) and PLC KO *C. perf* (right panel). CPE was the loading control. The size (kDa) of protein MW indicated. **K** HPLC analysis of DHA in WT and PLC KO *C. perf*. ($P = 0.0073, P = 0.0034$, two-way $t$ test, $n = 3$). Data are representative of three independent experiments as the mean ± SD (error bars). Source data are provided as a Source Data file.

compared with mice treated with PBS or scrambled microRNA as controls (Supplementary Fig. 8C). Taken together, these data suggest that GNV-RNA enhances PD-L1 immunotherapy against melanoma progression via gut microbial metabolites.

## GELN aly-miR159a-3p targeting bacterial phospholipase C (PLC) leads to accumulation of DHA

The fact that a dramatic induction of EPA and its metabolic product DHA in the SI of GELN-treated hFB mice was observed in the LC-MS/MS analysis but not in GF mice (Fig. 4A) further prompted us to investigate whether unsaturated fatty acids EPA and DHA contribute to the molecular mechanisms by which GELN-RNA enhances the anti-PD-L1 immunotherapy. Individual high-performance liquid chromatography (HPLC) analysis was utilized to verify the LC-MS analysis. As expected, the EPA and DHA were specifically induced by GELN but not the other PNP. This result was reproduced in hFB mice but not in GF mice (Fig. 7A). HPLC analysis also suggested that GELN-RNA encapsulated into GNV (GNV-RNA) dramatically induced DHA and EPA in hFB mice gut but not in the GF mouse gut (Fig. 7B). In contrast, a mixture of NV-RNA (NV-RNA$^{Mix}$) from garlic, aloe, and lemon had no influence on the EPA and DHA in gut feces and peripheral blood (Fig. 7B and Supplementary Fig. 9A). Given that *Lachnospirasceae* was one of the major targeted bacteria by GELN (Fig. 1A) mediated by DGDG of GELN (Fig. 3D), C. *perf*, a species in the *Lachnospirasceae* family was utilized to test our hypothesis that GELN-RNA uptake by *Lachnospirasceae* leads to induction of EPA and DHA that contributes to enhancing anti-PD-L1 immunotherapy. We examined DHA and EPA in the medium of *C. perf* using HPLC. We found that the DHA and EPA were induced by GNV-RNA but not by NV-RNA$^{Mix}$ (Fig. 7B and Supplementary Fig. 9A). We next sought to identify the mechanism underlying how GELN-RNA induces the production of DHA and EPA. Using our previously published GELN-RNA sequencing data in the NCBI GEO database (accession, GSE153126)[4,23], high abundance miRNAs sequences of GELN-RNAs (Fig. 7C) were input into the bacterial genomes database from the NCBI RefSeq database (ftp://ftp.ncbi.nlm.nih.gov/genomes/refseq/bacteria/) to search for potential targets of GELN miRNAs. An alignment of nucleotides sequences using BLAST indicated that GELN miRNA aly-miR159a-3p potentially targets phospholipase C (PLC) with two 7-mer lengths of reverse complement sequences (Fig. 7D), which contain a lipoxygenase domain that may catalyze DHA into 4-oxo-DHA and 4-HDHA[46,47] (Fig. 7E). To further address whether GELN aly-miR159a inhibits the expression of PLC in *Clostridiales*, we first confirmed using western blot analysis that PLC is expressed in *C. perf* and hFB mouse fecal microbes but not mouse intestinal tissue (Supplementary Fig. 9B). A chromatin immunoprecipitation (ChIP) assay also indicated that PLC DNA could be pulled-down by aly-miR159a-3p *C. perf* (Fig. 7F, **left panel**) and mouse feces (Fig. 7F, **right panel**) but not by aly-miR159a-

3p mutant. An analysis by qPCR (Fig. 7G) and western blot (Fig. 7H) generated from *C. perf* treated with aly-miR159a-3p indicated that the expression of the gene encoding PLC was indeed inhibited in transcription and at the translational level, respectively. HPLC analysis of bacterial medium indicated that aly-miR159a treatment leads to induction of DHA and reduction of 4-oxo-DHA but has no influence on EPA (Fig. 7I). This data suggested that GELN aly-miR159a-3p inhibits PLC expression. PLC can metabolize DHA into 4-oxo-DHA, therefore, inhibition of expression of PLC leads to DHA accumulated in *C. perf* (Fig. 7E). To further verify the role of PLC on DHA metabolism, the gene encoding for PLC was knocked out (KO) from the genome of *C. perf* (Fig. 7J) and replaced with an ampicillin-resistant sequence using fusion PCR-driven overlap extension[48,49]. PLC KO resulted in a significantly increasing level of DHA and a decreasing level of 4-oxo-DHA in *C. perf* (Fig. 7K). The aly-miR-159a-3p no longer influenced the level of DHA and 4-oxo-DHA in PLC deficient *C. perf* (Fig. 7K).

## DHA enhances anti-PD-L1 immunotherapy efficacy in melanoma

To further determine whether DHA plays a causative role in enhancing anti-PD-L1 therapy against melanoma progression, B16F10 melanoma-bearing mice were administered for one month PD-L1 Ab with/without DHA or EPA. The results indicated that both DHA and EPA improve the effectiveness of PD-L1 Ab against melanoma growth (Fig. 8A, top panel), lung metastasis (Fig. 8A, bottom panel), and prolonged survival of mice (Fig. 8B). Upon analysis of PD-L1 expression in tumor tissue using qPCR (Fig. 8C) and western blot (Fig. 8D), we found that both metabolites inhibited the expression of PD-L1 in mRNA and at the protein level, respectively. The B16F10 cells exposed to the DHA resulted in a decrease of PD-L1 expression in a DHA dose-dependent manner (Fig. 8E).

Given DHA inhibits PD-L1 expression and PD-L1 level represents the potential biomarker of response to anti-PD-L1 immunotherapy[50], we next investigate whether the levels of DHA and PD-L1 in clinical tumor samples were related to a therapeutic response. The expression of PD-L1 in 61 formalin-fixed paraffin-embedded archival melanoma specimens from PD-L1 responding ($n = 35$) and non-responding ($n = 26$) patients was analyzed. After PD-L1 immunohistochemistry (IHC) was performed, each sample was scanned and PD-L1 expression was scored using a computer-assisted method. The results suggested that the intensity of the PD-L1 signal (Fig. 9A, left IHC panel) and percentage of PD-L1$^+$ cells (Fig. 9A, right bar graph panel) apparently increased in melanoma tissue compared to adjacent normal tissue and tumor from immunotherapy non-responding patients exhibited higher expression of PD-L1 than the tumor tissue from responding patients. To further confirm these findings, the expression of PD-L1 in the samples was assessed using quantitative approaches. Consistently, the results from qPCR (Fig. 9B), ELISA (Fig. 9C), and immunoblot

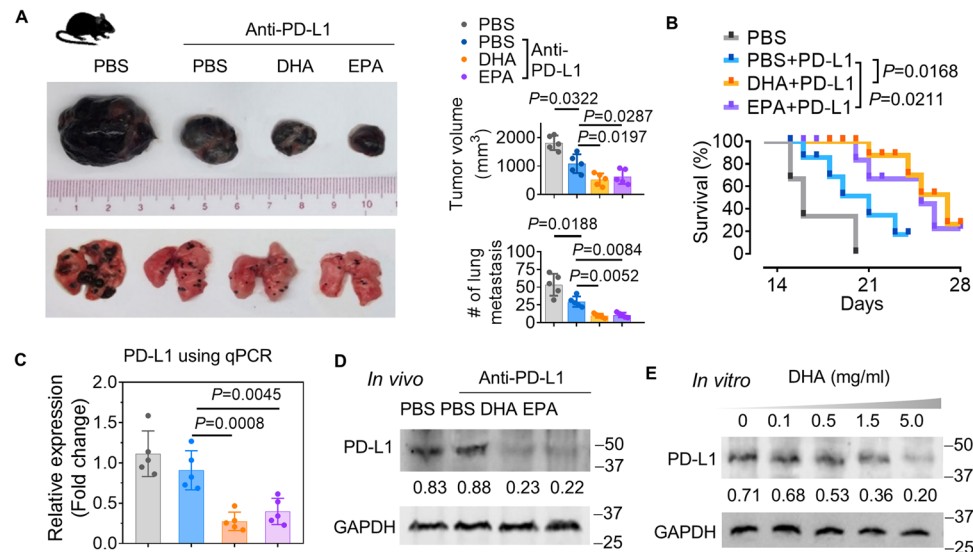

**Fig. 8 | DHA improves PD-L1 antibody effectiveness against melanoma mediated by reduction of PD-L1 expression in tumor cells. A** Representative B16F10 melanoma primary tumor (top left) and lung (bottom left) from tumor-bearing hFB mice (*n* = 5) at 28 d post-injection of B16F10 cells along with anti-PD-L1 antibody w/o DHA and EPA via oval gavage (50 mg/kg, body weight, every other day). Quantification of primary tumor volume (*P* = 0.0322, *P* = 0.0287, *P* = 0.0197, two-way ANOVA test, *n* = 5) and metastatic nodule number (>1 μm) (*P* = 0.0188, *P* = 0.0084, *P* = 0.0052, two-way ANOVA test, *n* = 5) (right panel). **B** Survival rate of mice in (A). (*P* = 0.0168, *P* = 0.0211, two-way Chi-Square test, *n* = 3). **C** PD-L1 analysis in tumor of

B16F10 melanoma xenograft mice in (A) using qPCR. **D** Representative immunoblots analysis of PD-L1 in tumor of B16F10 melanoma xenograft mice. GAPDH served as a loading control. The size (kDa) of protein MW indicated. Numbers below western blots represent densitometry values normalized to the loading control. **E** Western blot analysis of PD-L1 in B16F10 cells treated with DHA (0–5.0 mg/ml) for 2 h. The size (kDa) of protein MW indicated. Numbers below western blots represent densitometry values normalized to the loading control. Data are representative of three independent experiments as the mean ± SD (error bars). Source data are provided as a Source Data file.

(Fig. 9D) demonstrated that non-responding tumors exhibit higher PD-L1 levels compared to responding tumor, and non-responding tumor tissues have more PD-L1 mRNA and protein. In contrast, HPLC analysis suggested that DHA in tumors is decreased compared to adjacent non-tumor tissue, and DHA in tumors from non-responding subjects is lower compared to tumors from therapy-responding melanoma patients (Fig. 9E). This decrease in DHA was also found in the gut of subjects (Fig. 9F). Strong inverse correlations were observed between the level of PD-L1 and DHA (R² = 0.86–0.97).

To estimate the contribution of human gut bacteria in the immunotherapy response, the gut bacteria were isolated from the responding and non-responding subjects to PD-L1 therapy. PLC is required for metabolizing DHA into 4-oxo-DHA which contributes to the advancement and progression of cancer[47]. The expression of PLC was determined by qPCR and indicated that PLC in the bacteria from non-responding subjects is higher than in the bacteria from responding subjects (Fig. 9G). The bacteria from responding and non-responding subjects were then colonized in GF mice (Fig. 9H). We found that the commensal bacteria from responding subjects (Bac^Resp), but not the bacteria from non-responding subjects (Bac^Non) significantly improved the anti-PD-L1 immunotherapy by inhibiting tumor growth (Fig. 9I, top panel) and lung metastasis (Fig. 9I, bottom panel), as well as prolonged survival of the mice (Fig. 9J). The fact that Bac^Non supplied with DHA can enhance the immunotherapy efficiency in the same way as Bac^Resp does further suggests that DHA of gut bacteria plays a causative role in improving anti-PD-L1 immunotherapy via DHA mediated inhibition of PD-L1 expression in tumor cells. Given binding of PD-L1 to PD-1 interferes with the anti-tumor response and proliferation of T cells, especially CD8⁺ subpopulations[51], we next assessed the level of tumor-specific interferon-γ (IFNγ) in CD8⁺ tumor-infiltrating lymphocytes[52]. The CD8⁺PD-1⁺ cells were gated in FACS analysis (Supplementary Fig. 9C), and the result indicated that both Bac^Resp and DHA, but not Bac^Non dramatically promotes IFNγ production (Fig. 9K) and decreases apoptosis (Annexin V⁺7-AAD⁻) in CD8⁺PD-1⁺ T cells (Fig. 9L). Given natural and engineered PNPs are usually taken

up by local immune cells such as monocytes/macrophages[23,53], we tested whether GELNs target and influence immune response in tumor-infiltrating immune cells. GELNs labeled with the green fluorescence dye DiO were administered to B16F10 melanoma-bearing mice via oral gavage, and the colocalization of GELN/DiO and F4/80⁺ macrophage in the tumor was visualized by confocal microscope (Supplementary Fig. 9D). The results suggested that GELNs were taken up by tumor-infiltrating macrophages. FACS analysis further indicated that GELNs induced the IFNγ in GELN⁺F4/80⁺ cells but had little effect on the production of IFNγ in GELN⁻F4/80⁺ cells (Supplementary Fig. 9E).

Together, our data suggest that gut bacteria of anti-PD-L1 non-responding patients exhibits a high level of PLC activity. The high level of PLC activity in turn accelerates the metabolism of DHA and promotes PD-L1 expression in tumor, leading to the impairment of the anti-tumor activity and survival of T cells. Administration of Bac^Resp or DHA enhances the anti-tumor activity of tumor-infiltrating T cells and facilitates anti-PD-L1 immunotherapy efficacy in nonresponding mice.

## DHA inhibits the expression of PD-L1 via an epigenetic effect by binding DHA to the PD-L1 promoter region

Although we found that the bacterial metabolite DHA inhibits PD-L1 expression in tumor cells, resulting in the enhancing of anti-PD-L1 immunotherapy, how DHA inhibits PD-L1 expression remains unclear. A large body of evidence indicates that microbial-derived metabolites modulate the expression of host genes via epigenetic processes[54,55]. Whether plant-derived nanoparticles have an epigenetic effect on the mammalian host via gut bacterial metabolites released from PNP recipient bacteria has never been investigated. To address this question, we assumed that DHA modulates the PD-L1 expression by targeting the promoter region of PD-L1 since DHA regulates the expression of PD-L1 at both the mRNA and protein levels. To test whether DHA interacts with the promoter of PD-L1, we investigated the mobility of DNA oligos with or without DHA using single-strand gel shift (SSGS) methodology, which was recently developed[56]. Evidence of an interaction was based on an oligo mobility shift by DHA. The

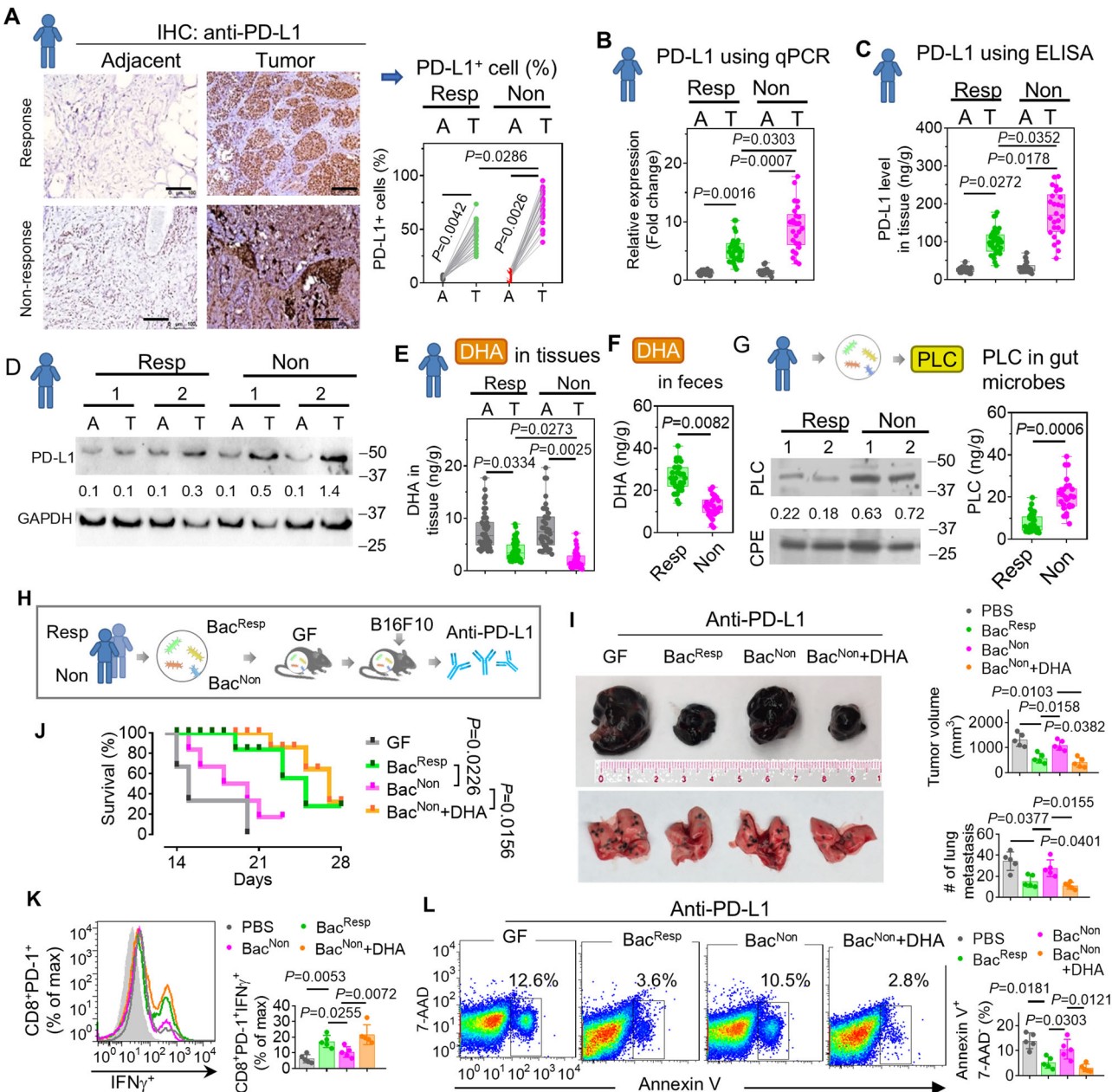

synthesized short single-DNA strains (around 60 mer in length each) whose sequences span the promoter of PD-L1 and overlap 10 mer each other (Supplementary Fig. 10Aand Supplementary Table 10). The site and sequence of the promoter and transcription start site (TSS) were determined according to the entire mRNA (Accession# GQ904196) and genomic sequence (Accession, NC000085.7 (11,441...24,385)) from the NCBI nucleic acid database. Twelve DNA oligos (PD-L1p1 to PD-L1p12, 2 μM/each) were incubated with DHA (10 μM) following nondenaturing polyacrylamide gel electrophoresis (PAGE) analysis (Fig. 10A). As expected, the SSGS assay indicated that only oligo PD-L1p9 (− 380 to − 321 upstream from the TSS) was mobility shifted by DHA but not EPA (Fig. 10B and Supplementary Fig. 10B), which indicates that DHA, but not EPA, binds to the promoter region of PD-L1. To define an accurate binding site of DHA to the promoter of PD-1L, we then tested the mobility shift using short oligos (PD-L1p9A to PD-L1p9F, 20 mer/each) covering the whole sequence of PD-L1p9 with overlapping sequences for each oligo (Supplementary Table 10). Only oligo PD-L1p9C displayed a shift (Fig. 10C), which indicated that the sequence of PD-L1p9C contained a potential essential sequence for

metabolite DHA binding. The oligo PD-L1p9C mobility shift was reestablished by gut supernatant from hFB colonization mice and even more from GELN-treated ones (Fig. 10D). The mutant of oligo PD-L1p9C abolished the mobility shift (Fig. 10D). To further verify the dynamic interaction of DHA and oligo PD-L1p9C, we next performed surface plasmon resonance (SPR) analysis, which is an optical technique for detecting the interaction of two different molecules in which a biotin-labeled oligo is immobilized on a streptavidin sensor chip while the DHA is brought into contact via the flow cell of an SPR instrument. The SPR assay results indicated that DHA binds to biotin-labeled (Bio-) PD-L1p9C, but not mutated (Bio-) PD-L1p9C (PD-L1p9C-Mut) (Fig. 10E). This result is consistent with the observation from the SSGS assay result. To further determine whether DHA binding to the PD-L1 promoter leads to altering the PD-L1 promoter activity, we introduced the mouse PD-L1 promoter sequence into a Gaussia luciferase reporter (pEZX-mPDL1, GeneCopoeia). The mutant (pEZX-mPDL1-Mut) with modified DHA binding sequences was used as a control. The luciferase assay demonstrated that DHA inhibits the luciferase activity in the cells transfected with pEZX-mPDL1 but not the mutant (Fig. 10F). To further

**Fig. 9 | Induction of PLC alleviates DHA level contributing anti-PD-L1 immunotherapy non-response in melanoma patients. A** Representative immunohistochemistry (IHC) analysis of PD-L1 level in PD-L1 antibody response (Resp, $n = 35$) or non-response (Non, $n = 26$) primary melanoma (T) specimens ($n = 20$) as well as their adjacent normal skin tissue (A) of patients (left panel). Scale bars, 300 μm. Quantification of PD-L1$^+$ cell ratio in the specimens (right panel). ($P = 0.0042$, $P = 0.0026$, $P = 0.0266$, two-way Chi-Square test). **B** PD-L1 expression in the tissues from the melanoma patients using qPCR. ($P = 0.0016$, $P = 0.0007$, $P = 0.0303$, two-way ANOVA test). **C** PD-L1 expression in the tissues from the melanoma patients using ELISA. ($P = 0.0272$, $P = 0.0178$, $P = 0.0352$, two-way ANOVA test). **D** Representative immunoblot analysis of PD-L1 in the tissues from the melanoma patients. The size (kDa) of protein MW indicated. Numbers below western blots represent densitometry values normalized to the loading control. **E** Analysis of DHA in melanoma of patients using HPLC. ($P = 0.0334$, $P = 0.0025$, $P = 0.0273$, two-way ANOVA test). **F** Feces from subjects suspended in PBS (0.5 g/ml) following centrifugation, the supernatant was used for DHA analysis with HPLC. ($P = 0.0082$, two-way $t$ test). **G** The bacteria isolated from the feces of melanoma patients and PLC level analysis using western blot (left panel) and ELISA (right panel). The size (kDa) of protein MW indicated. Numbers below western blots represent densitometry values normalized to the loading control. ($P = 0.0006$, two-way $t$ test). The box & whisker plots in Figs. above with the box representing the SD, the middle line within the box representing the median, and the bars representing the range of minimum and maximum. **H** Schematic representation of the administration to GF C57BL/6 J mice colonized with fecal bacteria from subjects of Resp and Non following B16F10 inoculation and PD-L1 antibody therapy ($n = 5$ mice). **I** Representative B16F10 melanoma primary tumor (top left) and lung (bottom left) from tumor-bearing mice ($n = 5$) at 28 d subjected to injection of B16F10 cells along with anti-PD-L1 antibody w/o DHA via oval gavage (50 mg/kg, body weight, every other day). Quantification of primary tumor volume and metastatic nodule number ($>1$ μm) (right panel). ($P = 0.0362$, $P = 0.0158$, $P = 0.0103$; $P = 0.0401$, $P = 0.0377$, $P = 0.0155$, two-way ANOVA test, $n = 5$). **J** Survival rate of colonized mice with B16F10 inoculation and administered bacteria and DHA by oral gavage. ($P = 0.0226$, $P = 0.0156$, two-way Chi-Square test, $n = 5$). **K** Monocytes isolated from the melanoma tissue. FACS analysis of IFNγ in CD8$^+$PD-1$^+$ cells (left panel). The strategy of gating used is the same as Supplementary Fig. 3C and Supplementary Fig. 9C. Quantification of IFNγ in CD8$^+$PD-1$^+$ cells$^-$ (right panel). ($P = 0.0053$, $P = 0.0255$, $P = 0.0072$, two-way Chi-Square test, $n = 5$). **L** Monocytes isolated from the tumor. FACS analysis of apoptosis by flow cytometry using Annexin V-FITC staining in CD8$^+$PD-1$^+$ cells of tumor from mice. Numbers in boxes indicate a representative percent of CD8$^+$PD-1$^+$ apoptotic cells (Annexin V$^+$7-AAD $^-$) (left panel). Quantification of the percentage of apoptotic CD8$^+$PD-1$^+$ cells (right panel). ($P = 0.0181$, $P = 0.0303$, $P = 0.0121$, two-way Chi-Square test, $n = 5$). Data are representative of three independent experiments as the mean ± SD (error bars). Source data are provided as a Source Data file.

reveal the specific DNA motif necessary for DHA binding, we synthesized 20 mer length DNA oligos that contained the potential DHA binding motif. Among the 20 oligos, each oligo contained one mutated base. We assumed that the DHA binding activity of an oligo would diminish if an essential base was replaced. As expected, the SSGS analysis suggested that the sequence AGCCT......ATCAGC, − 341 to − 358 upstream of the PD-L1 TSS was necessary for DHA binding (Fig. 10G). Multiple sequence alignment indicated that this PD-L1 binding motif is a conserved sequence in humans and mice with both sharing this consensus sequence in the promoter region of the PD-L1 gene (Fig. 10H). The interaction of DHA with an oligo containing the human PD-L1 promoter region binding sequence was confirmed with the SSGS assay (Supplementary Fig. 10C).

More interesting, the prediction of transcription factor binding sites using PROMO 3.0 (http://alggen.lsi.upc.edu) indicated that both mouse and human PD-L1 promoter have a c-myc binding motif (CCAGGTG) near the DHA binding motif (Fig. 10Hand Supplementary Fig. 10A). To determine the effect of DHA on c-myc mediated PD-L1 expression, the c-myc gene was knocked out in B16F10 melanoma cells by transfection of c-myc CRISPR/Cas9 KO plasmid (Santa Cruz, sc-421770). Three days after transfection, the abolishment of expression of c-myc was confirmed by western blot (Supplementary Fig. 10D). The luciferase assay using the reporters containing mouse (pEZX-mPDL1) and human (pEZX-hPDL1) PD-L1 promoter sequence demonstrated that c-myc deficiency canceled the DHA-mediated inhibition of the luciferase activity in the cells transfected with pEZX-mPDL1 and pEZX-hPDL1 (Fig. 10I and Supplementary Fig. 10E). Co-transfection of recombinant c-myc with pEZX-mPDL1 or pEZX-hPDL1 restored DHA inhibitory activity. This data suggests that bacterial metabolite DHA enhances the efficacy of anti-PD-L1 immunotherapy by inhibition of c-myc mediated induction of PD-L1 expression in tumor cells.

To further identify the site and structure of molecular binding between DHA and PD-L1 promoter, the secondary structures of mouse and human PD-L1 promoter sequence containing DHA targeting sequences (Fig. 10H) were tested using Mfold[57] followed by analysis of docking using the HDOCK Server (http://hdock.phys.hust.edu.cn/). The potential mode of molecular interaction analyzed with UCSF Chimera[58] suggested the interaction between DHA and PD-L1 promoter via hydrogen bonds at position G-4 (m) and G-11 (h) and takes place with a docking score of -39.98 and ligand rmsd (Å) of 19.83 (Fig. 10J). Additional analysis of the SPR demonstrated that DHA inhibits c-myc recruiting to the promoter of PD-L1 as a transcription factor (Fig. 10K). The result of DHA interaction with the PD-L1 promoter

region generated from test tube experiments was further demonstrated in live cells. B16F10 melanoma cells were treated with gut fecal supernatant and biotin-labeled (Bio)-PD-L1p9C (Bio-WT) or Bio-S100p1-G mutant (Bio-Mutant) for 6 h and the complex was pulled-down from the cell lysate with streptavidin beads for quantitative analysis of metabolites. The LC-MS results suggest that DHA was recruited to the complex by the PD-L1p9C but not by the mutant (Fig. 10L).

Taken together, these data demonstrate that DHA inhibits the expression of mouse and human PD-L1 in melanoma cells through an epigenetic effect. The biological effect of DHA enhancing anti-PD-L1 therapy was further indicated by the fact that gut DHA levels and PD-L1 expression exhibited a negative correlation to each other.

## Discussion

Extensive research in the field of gut microbiome science has underscored the significant role of diet in shaping the species and functions of gut bacteria[59]. Diet exerts rapid influence on the composition and function of metabolites released by gut microbiota[4,60], which in turn interact with the host. Due to the importance of diet and the gut microbiota axis to human health[61], it is of critical importance to gain a mechanistic understanding of how diet-related factors regulate gut microbiota activity. A growing body of research has revealed that diet-derived factors are not just passive residents in the gut, but they play an important role in modulating vital gut microbiota activity and host physiological systems, both locally and systemically[3]. Understanding how microbes interact with diet-derived factors is therefore not only biologically interesting, but it provides a platform for discovering potential therapeutic targets for a variety of human diseases. However, our diet contains numerous unaccounted for factors that can modulate the functions of gut microbiota. A huge challenge in gut microbiome science is identifying how specific diet-derived factors influence gut microbiota that can modulate the health status of humans. Moreover, decades of microbiome studies using a combination of multi-omics approaches, have revealed strong associations between diet, microbiota, and human health[62]. Identifying the causal molecular mechanisms behind these associations is challenging: Which microbes, genes, and their metabolites regulated by diet-derived factor(s) positively and negatively, if any, are responsible for a host phenotype? And if these diet-derived factor(s) do affect us, how can we determine which diet-derived factor(s) have a role in promoting gut health via targeting gut microbiota? In the past years, the numbers and quality of PNP-related studies have dramatically increased[63-68].

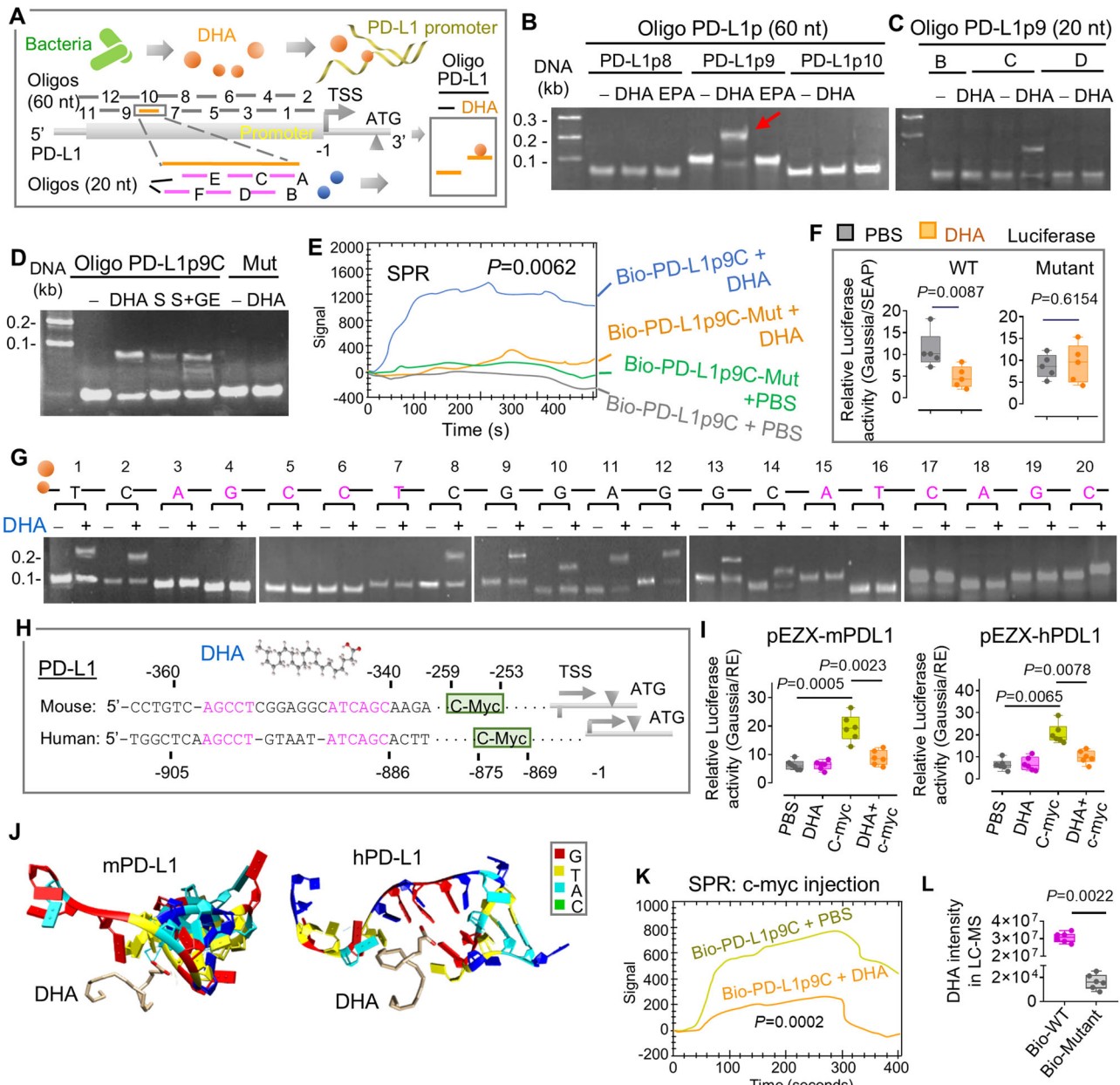

**Fig. 10 | DHA interferes with transcription factor c-myc access to the PD-L1 promoter. A** Schematic diagram of the strategy to demonstrate the interaction of DHA and the promoter of PD-L1 (left panel). Oligo binding to DHA with the expected mobility shift on PAGE (SSGS, right panel). The transcription start site (TSS) is marked by the bent arrow. ATG; translation start code. **B** 10 pmol of synthetic DNA oligo PD-L1p8, PD-L1p9, and PD-L1p10 (60 mer/each) corresponding to the sequence on the promoter of PD-L1 incubated with DHA (1 μM) or EPA (1 μM) for 30 min at 37 °C. The oligos separated on 15% native PAGE and visualized with ethidium bromide. The red arrow indicates oligo migration shifted. **C** The shorter synthetic DNA oligos PD-L1p9A to PD-L1p9F (20 mer/each) correspond to the sequence of PD-L1p9. Representative PAGE for the oligos PD-L1p9B to PD-L1p9D with or without DHA. **D** Oligo PD-L1p9C incubated with gut supernatants (S) from hFB colonization mice without or with GELN treatment (S + GE). Representative SSGS showed the mobility shift of oligo PD-L1p9C and oligo mutant. Oligo PD-L1p9C mutant was used as the control. **E** Surface plasmon resonance (SPR) analysis of the interaction of biotin-labeled oligos PD-L1p9C and mutant PD-L1p9C-Mut with DHA (1 μM.) (P = 0.0062, two-way t test, n = 5). **F** The promoter sequences of PD-L1 and mutant inserted into a luciferase reporter p pEZX and transfection of microglia. Luciferase activity assessment 12 h after treatment with DHA (P = 0.0087, P = 0.6154, two-way t test, n = 5). **G** Representation of a 15% PAGE for the oligo PD-L1p9C, as well as mutants indicated in the figure. Each oligo contained a single base

mutation. The size (kb) of the DNA length is indicated near the figure. The base in pink replaced by a different base, caused the abolishment of the DHA binding shift. **H** The sequence of mouse and human PD-L1 promoter indicating the distance from the TSS containing the DHA potential binding motif (pink) and c-myc binding site (box). **I** The analysis of luciferase activity for the c-myc KO B16F10 cells transfected with the luciferase plasmid containing mouse (m) and human (h) PD-L1 promoter sequence (P = 0.0005, P = 0.0023; P = 0.0065, P = 0.0078, two-way ANOVA test, n = 6). Treatment of DHA and/or recombinant c-myc protein is indicated in the figure. **J** 3D predicted structures of the interaction between DHA and oligo mPD-L1p9C or hPD-L1p (G- Red; T- Yellow; C- Green; A- Cyan) at position G-4 and G-11 by hydrogen bond respectively. **K** SPR analysis of the interaction between c-myc protein and biotinylated oligo PD-L1p9C covalently immobilized onto the sensor chip with or without DHA (P = 0.0002, two-way t test, n = 3). **L** Biotinylated oligo PD-L1p9C (WT) and mutant transfected into microglia for 12 h and incubated with gut supernatant for an additional 6 h. Metabolite analysis with LC-MS after pull-down with streptavidin beads (P = 0.0022, two-way t test, n = 6). Data are representative of three independent experiments. The box & whisker plots with the box representing the SD, the middle line within the box representing the median, and the bars representing the range of minimum and maximum. Source data are provided as a Source Data file.

Multiple-omics methodologies were used in this study to assess if oral administration of PNP influenced the human gut microbiome and microbiome-derived metabolites in distinct ways.

We employed multiple-omics methodologies to investigate the impact of PNP on the human gut microbiome and microbiome-derived metabolites. Our findings reveal that PNPs lipid and amino acid content act as "eat-me" and "don't-eat-me" signals, targeting specific gut bacterial species and their functions. While such signaling is essential in maintaining host homeostasis in mammals[6,69,70], its presence in diet-derived nanoparticles for bacteria remains understudied. We observed that PNP is selectively taken up by gut microbiota, with uptake depending on the lipidomic and amino acid profiles of PNP. Understanding the ratios of lipid and amino acid types in PNP may provide insights into their uptake by specific gut bacteria and subsequent metabolite production. Certain gut bacteria preferentially take up PNP containing specific lipids or amino acids, influencing their metabolic activities, and consequently affecting human metabolic pathways. This knowledge could be instrumental in targeted manipulation of the gut microbiome for precision medicine applications. Furthermore, our study suggests that the interaction between PNP and specific gut bacteria may contribute to regulating gut microbiome homeostasis, with implications for both healthy and unhealthy dietary patterns.

The past few decades of research in the microbiome field have made it evident that human biology is explicitly linked to microorganisms, with the majority of the microorganisms living in the digestive tract, where they produce or modify various chemicals, or trigger host reactions that affect various physiological functions including immunity, neurobiology and metabolism[43,71]. When considering the translational implications of microbiome research, it is evident that diet influences the microbiome, transcriptome, and metabolome profiles of the host. However, very little of the novel knowledge has matured to stages where it can be translated into clinical practice. To realize the application and clinical use of this new knowledge, analytical data on the microbiome must be integrated with other omic readouts and include omics generated from a healthy diet and modifications of a healthy diet. Knowledge about the diet-derived factor(s) that regulate transcriptional and translational activities of the gut microbiome is sparse and affects our ability to correctly interpret the outcome of metabolome profiling. In this study, using multi-omics profiling analytical approaches, we demonstrated that upon PNP entry into targeted bacteria, the metabolome can be positively altered in the targeted bacteria. Then, using ginger PNP as an example, we demonstrated how a PNP can alter the targeted bacterial metabolome via miRNA-mediated inhibition of bacterial gene expression. Specifically, we observed that gut metabolite DHA is accumulated in the gut. DHA accumulation is caused by ginger ELN-derived aly-miR159a mediated inhibition of the expression of the bacterial PLC gene. PLC is required for metabolizing DHA into 4-oxo-DHA which contributes to the advancement and progression of cancer[47]. DHA and EPA are two key omega-3 fatty acids and are mainly found in fish, plant seeds microbes, etc. The benefits of DHA and EPA have been observed for myocardial infarction[72], lowering triglycerides, insulin resistance[73], inflammatory responses[74], thrombosis and cognitive decline[75], however, mammals cannot make DHA and EPA due to the inefficiency in the conversion of α-linolenic acid (ALA) to DHA and EPA. The gut microbiome plays a critical role in the metabolism of DHA and EPA[76]. Emerging evidence indicates that the serum levels of DHA and total omega-3 fatty acids are significantly correlated with gut microbiome composition, particularly with bacteria of the *Lachnospiraceae* family[77]. Our studies suggest that edible plant-derived PNP increase the level of DHA by modulating the DHA metabolic pathway of gut microbes. Compared to a single omics approach, a multi-omics approach, as taken in this study, provides a more comprehensive, overarching, and complete picture of how plant lipids, amino acids, and RNA of a PNP react and induce/prevent particular metabolic actions within the gut microbiome

Finally, our study sheds light on the role of diet-derived factors in epigenetic modulation mediated by the gut microbiota. The alterations in function and composition of the gut microbiota have been known to be involved in the pathogenesis of metabolic diseases via induction of epigenetic modulation such as DNA methylation, histone modifications, and regulation by noncoding RNAs[78,79]. These induced epigenetic modifications can be regulated by metabolites produced by the gut microbiota, including short-chain fatty acids[80], USF[81], folates, biotin, and trimethylamine-N-oxide[82]. However, whether healthy diet-derived factors can cause epigenetic changes is not known. In this study, we demonstrate how DHA, a gut bacterial metabolite, can regulate PD-L1 expression, suggesting a novel avenue for studying the interactions between PNP, the gut microbiome, and epigenetic processes in human health and disease prevention.

Although in this study we demonstrated the role of ELN and Nano10, we cannot exclude the possibility that other types of PNP may also be present in the edible plants and work with ELN and Nano10 as a team for maintaining the homeostasis of host cells and tissues.

The pathways that regulate the homeostasis of host cells and tissues could be regulated by edible PNP in a positive and negative manner. In this study, a heat map (Fig. 4C) indicates that ELN and Nano10 showed opposite activities on pathways such as D-Glutamine and D-Glutamate (DGG), Arginine and Proline (ARP), as well as Taurine and Hypotaurine (THT). Since the composition of ELN is different from Nano10, quantitatively and qualitatively, the metabolites released from ELN-positive gut bacteria are different from Nano10-positive bacteria. Therefore, it is conceivable that their biological effects on the pathways in the human host cells will be different or even have opposite effects. This finding provides a rationale for further investigating whether PNP mediated opposite effects on these pathways may contribute to regulating host cell homeostasis.

The gut microbiota, including bacteria, archaea, fungi, and viruses, compose a diverse mammalian gut environment and are highly associated with host health. In this study, we demonstrate that PNP can be selectively uptake by gut bacteria. This finding opens up a new avenue to further study whether PNP can have effects on other members of the gut microbiota. Moreover, we demonstrated that lipid nanoparticles made from the total lipids extracted from GELN can deliver aly-miR-159a to suppress the expression of the recipient bacterial PLC gene, which reprograms the bacterial unsaturated fatty acids metabolism. This strategy has several advantages over other systems like lipid nanoparticles for delivery of the miRNA. First, unlike other delivery systems, GELN-derived lipid nanoparticles are made from natural healthy diet-derived material, and it is unlikely to have significant side effects to be considered; second, the byproducts generated from GELN-derived nanoparticles have the minimum potential for environment contamination, and other factors in/on the GELN derived lipid nanoparticles may also contribute to enhancing miR mediated therapeutic effect.

Our data presented in this study show that higher PD-L1 expression is associated with nonresponding to anti-PD-L1 immunotherapy. Administration of DHA enhances the anti-tumor activity and facilitates anti-PD-L1 immunotherapy efficacy in nonresponding mice. The tumor-immune interaction can be arbitrarily divided into seven steps: the release of cancer antigens, cancer antigen presentation, priming, and activation, trafficking of T cells to tumors, infiltration of T cells into tumors, recognition of cancer cells by T cells, and killing of cancer cells. Given that PD-1/PD-L1 blockade is primarily involved in the last step, any abnormalities in the previous steps may affect the efficacy of PD-L1 therapy and lead to resistance[83,84]. Abnormally upregulated PD-L1 expression and a lack of PD-L1 can both lead to the inefficacy of PD-L1 therapy[85,86]. Also, in the anti-tumor immune process, there are some other immune checkpoints, including CTLA-4, TIM3, LAG3, NKG2A, and so on, which are also recognized and participate in co-regulating immune responses with anti-PD-L1 therapy. Our data provides a

rationale for further investigating whether DHA also has a role in modulating the activity of the other six steps during tumor-immune system interaction.

Zhang et al.[87], in vitro data show that the PD-L1 expression in A549, LLC, HepG2, and SMMC-7721 cells was not affected by DHA treatment the transcriptional level at a dose up to 150 μM, which is equal to 0.05 mg/ml according to the molecular weight of DHA (328.488). In contrast, we treated the B16F10 cell with concentrations from 0.1 mg/ml to 5.0 mg/ml. DHA-mediated inhibition of PD-L1 protein expression was further supported by the three independent western blots (Fig. 6E) and the data suggests that the DHA significantly reduced PD-L1 expression in B16F10 at the concentration at 0.5 mg/ml and above. In addition, 250–1000 mg of DHA be consumed daily for most healthy adults without any side effects[88]. According to FDA human equivalent dose (HED) draft guidelines[89], for a 60 kg of human body, the HED of DHA for a mouse is 51–200 mg/kg. In our in vivo studies, mice were administered 50 mg/kg/d via oral gavage.

In the past decades, diet-derived materials, including proteins and lipids, were considered to be degraded in the gastrointestinal tract. However, when these materials are packed in vesicles such as exosomes, they remain stable[15]. Plant-derived ELNs have similar components as mammalian exosomes and are stable in vivo via oral administration[4,16–18]. In this study, we demonstrated that Nano10 is also stable in the gut. Whether this finding applies to other unidentified diet-derived nanoparticles needs further investigation.

It is important to realize that, compared to the free form of individual factors, PNP-associated individual factors gain new roles through interactions with neighboring factors encapsulated in the PNPs. We have demonstrated that these interactions occur within the exosomes, not in the exosome's donor cells because the proximity of factors for interaction is closer within exosomes than within the cytosol of donor cells[90]. The new roles of PNP-associated factors could have effects on the stability of PNP in the gut, the specificity of gut bacterial-mediated uptake, and the biological effects of PNPs on recipient cells.

PNP stability could be affected by the interaction of neighboring factors within the PNP. For example, a lipid bilayer consisting of multiple lipids protects their contents from the harsh environment of the gut[91,92]. Most of the diet-derived material in the gut is degraded via an enzymatic reaction that occurs by recognition of the structure of the diet-derived material. In comparison with the free form of these materials, when recognition domains of these materials is buried within the diet-derived nanoparticles, recognition for degrading is improbable or less likely. These mechanisms collectively aid edible PNP in maintaining their integrity and function within the gastrointestinal tract.

It is conceivable that not only does each individual factor contribute to the PNP stability, it also plays a role in the specificity of uptake by gut bacteria by affecting the PNP size, charge, and shape. Smaller nanoparticles are generally taken up more efficiently due to their ability to penetrate bacterial cell walls more easily[93]. The shape of the nanoparticles also plays a role, with different shapes interacting differently with bacterial surfaces[94]. The surface charge of nanoparticles affects their interaction with bacterial cell membranes. Positively charged nanoparticles tend to interact more strongly with negatively charged bacterial cell walls[95]. Moreover, the dynamic structure of PNP may also undergo conformational changes to perform their functions[96]. These dynamic changes are essential for processes like PNP enzyme catalysis, signal transduction, and molecular recognition. Understanding these factors can help in selecting specific PNP for specific applications, such as targeted drug delivery or antibacterial treatment.

In this study, PNP is referred to as nanoparticles isolated from edible plants. As a proof of concept, Nano10 and ELN were isolated and studied. Nano10 is derived from the pellet after a $10,000 \times g$

centrifugation, while ELN is derived from the pellet after a $100,000 \times g$ centrifugation. Unlike mammalian-cell-derived exosomes, which are spontaneously released from mammalian cells, Nano10 and ELN are not spontaneously released from plant cells. Instead, they are released when the edible plant cell is broken down, such as during chewing and digestion. On the other hand, ELN has similar components, including proteins, lipids, and RNA, as mammalian cell exosomes. We expect that in addition to Nano10 and ELN, other types of PNP are present in the plant diet.

## Methods

### Mice

Six- to eight-week-old male specific-pathogen-free (SPF) C57BL/6 J mice (stock, 000664) were purchased from the Jackson Laboratory (Bar Harbor, ME). All mice were housed under SPF conditions and supplied with a regular chow diet (LabDiet, 5001). 6–8 weeks old male germ-free (GF) inbred C57BL/6 J mice (stock, N000295) were purchased from the National Gnotobiotic Rodent Resource Center (University of North Carolina, NC, Chapel Hill, NC) and maintained in flexible film isolators (Taconic Farm) at the Clean Mouse Facility of the University of Louisville. Animal care was performed following the Institute for Laboratory Animal Research (ILAR) guidelines, and all animal experiments were conducted in accordance with protocols approved by the University of Louisville Institutional Animal Care and Use Committee (IACUC, Louisville, KY). The mice were acclimated for at least 1 week before any experiments were conducted. The maximum tumor size for a mouse is $2000 \text{ mm}^3$ according to standard guidelines[97]. The volume of the tumor was evaluated by caliper measurements using the formula: volume = (width)$^2$ × length/2[98]. The mice were monitored twice daily and mice bearing a tumor size exceeding 20 mm in any direction were euthanized. In addition, mice having difficulty in walking or losing their balance, and losing 10% of their body weight over one week were euthanized using carbon dioxide, supplied from a cylinder or tank in a chamber fitted with an appropriate pressure-reducing regulator and flow meter to ensure a gradual displacement of 30–70% volume/minute of the euthanasia chamber. $CO_2$ flow was maintained for at least 1 min after respiratory arrest, and death was ensured by an adjunctive physical method such as cervical dislocation.

### Human subjects

The study cohort involved (1) 24 healthy volunteers (from 25 to 52-year-old men). All human fecal samples from healthy volunteers were collected for bacteria isolation in the Department of Surgery, Huai'an First People's Hospital, Huai'an, Jiangsu, China after written informed consent from patients. Volunteers were recruited from the population in 2022 in Huai'an, Jiangsu, China. No subjects had a history of chronic gastrointestinal disease, taking antibiotics within three months of testing, alcohol abuse, or smoking. (2) 61 melanoma patients (men) were assigned to a PD-L1 responding group ($n = 35$) and a non-responding group ($n = 26$). Anti−PD-L1 antibody was administered (1–5 mg per kilogram of body weight) every 14 days in 6-week cycles for up to 16 cycles or until the patient had a complete response or confirmed disease progression[99]. Objective responses were confirmed by at least one sequential tumor assessment[99]. All clinical samples, including tissue and fecal samples, were collected in the Department of Surgery, Huai'an First People's Hospital, Huai'an, Jiangsu, China, after written informed consent from patients. Approval for the study was granted by the Institute Research Ethics Committee at the Health Department of Huai'an.

### Cells

C57BL/6 murine melanoma B16F10 cells (CRL-6475, American Type Culture Collection, ATCC) were grown in tissue culture plates with Dulbecco's modified Eagle's medium (DMEM) supplemented with

10% heat-inactivated fetal bovine serum (FBS), 100 μ/mL penicillin, and 100 μg/mL streptomycin at 37 °C in a 5% CO₂ atmosphere.

### Bacteria
*Clostridium perfringens* (*C. perf*, ATCC 13124, Manassas, VA) was grown in reinforced clostridial medium (RCM) (Thermo Sci., OXCM0149B) and *Lactobacillus rhamnosus* (LGG, ATCC 53103, Manassas, VA) was grown in De Man, Rogosa and Sharpe (MRS) broth (Hardy Diagnostics, Santa Maria, CA) at 37 °C in anaerobic conditions. Cultures were centrifuged, and the bacterial pellet was diluted in the medium for in vitro experiments and in PBS for gavaging at $10^9$ colony-forming units (CFU)/mouse per day.

### Isolation of bacteria from human feces
Human fecal samples were collected in a sterile and anaerobic container (BD GasPak EX Gas Generating Container system with BD Gas Generating Pouch system with indicator) on ice[100]. The fecal samples were homogenized in sterile phosphate-buffered saline (PBS) (100 mg/ml) and centrifuged at $1000 \times g$ for 5 min at 4 °C. The supernatant was passed through a metal strainer and centrifuged at $6000 \times g$ for 15 min at 4 °C[101]. The bacterial pellet was resuspended in sterile PBS, and the bacterial number was counted using a Petroff-Hausser chamber. An equal volume of 20% sterile glycerol was mixed with bacteria in PBS for storage at − 80 °C until use.

### Bacterial colonization
Six- to eight-week-old male GF mice ($n = 15$) were colonized by intragastric gavage with $1 \times 10^9$ colony forming unit (CFU) and rectal inoculation with $2 \times 10^9$ CFU of fresh human feces-derived bacteria (hFB)[31–33]. Inoculation was repeated 72 h after the initial inoculation. Mice were subsequently housed in sterile conditions. The bacterial colonization of the mice was evaluated once a week throughout the experiments. Over multiple generations, the hFB-colonized mice were used for further experiments.

### Preparation of plant-derived nanoparticles
To prepare plant nanoparticles (PNP), peeled Hawaiian ginger roots (Simply ginger, PLU:4612), garlic (PLU:3399), aloe (PLU:3064), and lemon (PLU:3362) were used for isolation and purification of nanoparticles using a previously described method[4]. Briefly, the plants listed above were peeled after disinfecting with ethyl alcohol (70%) and then homogenized in a high-speed blender for 1 min. The juice was collected after net filtration and diluted with sterile PBS. The supernatant was collected after centrifugation at $1000 \times g$ for 10 min, $2000 \times g$ for 20 min, $4000 \times g$ for 30 min, and $10,000 \times g$ for 1 h. The pellets collected at $10,000 \times g$ were denoted as Nano10. The pellets containing nanoparticles derived from each plant were spun down at $100,000 \times g$ for 1.5 h at 4 °C and denoted as exosome-like nanoparticles (ELN). The isolated nanoparticles were further purified in a sucrose gradient (8, 30, 45, and 60% sucrose in 20 mM Tri-Cl, pH 7.2) centrifugation at $100,000 \times g$ for 1.5 h at 4 °C. All containers and reagents used were kept bacteria-free by autoclaving or 0.22 μM filtration. Purified ELN and Nano10 were characterized by morphology with a Zeiss EM 900 electron microscope using a previously described method[102]. Size distribution and concentration were determined using a NanoSight NS300 (Malvern Instrument, UK) at a flow rate of 0.03 ml per min.

### Extraction of PNP lipids
To isolate the lipids from purified PNP, 1-part volume of PNP was mixed with 1.25-parts chloroform and 2.5-parts methanol in a glass tube and mixed well. An additional 1.25-parts chloroform and 1.25-parts H₂O were added and mixed well. The mixture was centrifuged at 2000 rpm for 10 min at room temperature, and the lower organic phase as

removed and dispensed into a new clean glass tube[103]. The collections and lipids were dried by nitrogen (2 psi) and stored at − 80 °C.

### Lipidomic analysis with liquid chromatography-mass spectrometry (LC-MS)
Lipid samples extracted from PNP were submitted to the Lipidomics Research Center, Kansas State University (Manhattan, KS) for analysis using a method previously described[104]. In brief, the lipid composition was determined using triple quadrupole MS (Applied Biosystems Q-TRAP, Applied Biosystems, Foster City, CA). The protocol has been previously described[104]. The data are reported as the concentration (nmol/mg PNP) and percentage of each lipid within the total signal for the molecular species determined after normalization of the signals to internal standards of the same lipid class.

### Generation of Nanovesicles (NV) with PNP lipids
To generate NV, 1 mg of lipid was incubated with 200–400 μl of PBS. The NV was generated with/without ampicillin with ultrasound in a bath sonication (FS60 bath sonicator, Fisher Scientific) for 30 min[4]. The pelleted particles were collected by centrifugation at $100,000 \times g$ for 1 h at 4 °C. The size distribution and concentration of the particles were analyzed with nanoparticle tracking analysis (NTA) using a NanoSight NS3000. Purified NVs were fixed in 2% paraformaldehyde and imaged under a Zeiss EM 900 electron microscope.

### PNP protein extraction
To disrupt the complex polysaccharide plant cell wall, ionic detergent cetyltrimethylammonium bromide (CTAB) was used to break down the membranes of PNP[105]. 100 mg of PNP were vortexed with CTAB lysis buffer (2% CTAB, 2% polyvinylpyrrolidone (PVP), 2.0 M NaCl, 20 mM EDTA, 100 mM Tris-HCL (pH 8.0) and 5% ß-mercaptoethanol) following incubation at 60 °C for 30 min. After centrifugation at $20,000 \times g$ for 10 min, the supernatant of the protein extract was collected and the concentration of protein was measured using a BCA Protein Assay kit (Thermo Fisher Sci., A55865).

### PNP membrane amino acids extraction
PNP was disrupted with CTAB buffer. After centrifugation at $20,000 \times g$ for 10 min, the pellet containing the PNP membrane was mixed with chloroform/methanol/H₂O solvents. The lipids are dissolved in the organic phase and the hydrophilic phase was collected for amino acid analysis using LC-MS analysis.

### Labeling of nanoparticles and bacteria with fluorescent dye
Plant nanoparticles were labeled with PKH26 Fluorescent Cell Linker Kits (Sigma, PKH26GL) in accordance with the manufacturer's instructions. After washing with PBS, bacteria pellets, PNP or PNP nanovectors were suspended in 250–500 μl of diluent C with 2–4 μl of PKH26 and subsequently incubated for 30 min at room temperature. After centrifugation for 5 min at $13,000 \times g$, labeled nanoparticles were resuspended for further experiments.

### Bacteria uptake of PNP Assay
To identify the nanoparticles taken up by bacteria in vitro, $1 \times 10^8$ human fecal bacteria were incubated with 1 mg of PKH26-labeled ELN or Nano10 in PBS for 30 min at room temperature. After two washes with PBS, the bacteria suspension was loaded on a slide and the uptake of nanoparticles was visualized using confocal microscopy. To exclude the possibility of detecting ELN or Nano10 remaining on the outside of bacteria, the bacteria were washed three times with medium and treated with 100 μl of 0.5% Triton X-100 for eight minutes. Bacteria broth was immediately added to wash the bacteria 2x before the bacteria were imaged using confocal microscopy. (Note: 0.5% Triton X-100 did not affect bacterial viability for at least 30 min after addition). To identify the nanoparticles taken up by bacteria in vivo,

PKH26-labeled PNP was administered to hFB colonized GF C57BL/6 J mice (500 mg/kg, body weight) via oral gavage. Two hours after administration, the gut fecal bacteria were isolated and PKH26+ bacteria were quantified and sorted using a BD FACSAria™ III cell sorter (BD Biosciences, San Jose, CA).

## Microbiome analysis with 16S rRNA gene sequencing
Microbial genomic DNA (gDNA) from fecal samples was isolated with QIAamp DNA Stool Mini Kits (Qiagen, Cat 51504) following the manufacturer's instructions, and bacterial strains were investigated using 16S rRNA gene sequencing. DNA (15 ng) was used as a template to amplify the 16S rRNA gene using a high-fidelity PCR System kit (Roche, Cat 03310256103). The v1-v3 regions of the 16S ribosomal RNA gene were amplified using 27 f (AGAGTTTGATCCTGGCTCAG) and 534r (ATTACCGCGGCTGCTGG) primers (1 μM). The primers were anchored with adapter (adapter A: 5′ CCATCTCATCCCTGCGTGTCTCCGACT-CAG 3′ and adapter B: 5′ CCTATCCCCTGTGTGCCTTGGCAGTCTCAG 3′) and Multiplex Identifiers (MIDs; 10 bp long). The multiplexed amplicons were purified using a QIAquick Gel Extraction Kit (Qiagen, Cat 28704). The amplicon sequence was conducted using the 454 Jr. Sequencing platform. The 16S rRNA gene sequences were analyzed using QIIME 2 platform scripts (www.qiime.org). The microbial classification was performed with the 16S rRNA reference SILVA databases using QIIME 2 tools[106,107]. By applying hierarchical clustering algorithms (HCA), we determined the species clustering based on the operational taxonomic unit (OTU) using amplicon sequencing of 16S RNA. The reference sequences allowed the sorting of the results into OTUs by clustering 97% sequence similarity (uclust) and classification according to various taxonomic ranks (phylum, order, class, family, genus, and species). The percentage of each bacterial species was virtualized with R software.

## Metabolomics analysis with LC-MS
Fecal samples were collected and suspended with PBS. The supernatant was collected after centrifugation at $6000 \times g$ for metabolomics analysis using 2D-LC-MS/MS in positive mode and negative mode to acquire MS/MS spectra for metabolite identification[108]. MetSign software was used for spectrum deconvolution, metabolite identification, cross-sample peak list alignment, normalization, and statistical analysis. To identify metabolites, the 2D-LC-MS/MS data of pooled samples were first matched to our in-house MS/MS database that contains the parent ion m/z, MS/MS spectra, and retention time of 187 metabolite standards. The thresholds used for metabolite identification were MS/MS spectral similarity ≥ 0.4, retention time difference ≤ 0.15 min, and m/z variation ≤ 4 ppm. The 2D-LC-MS/MS data without a match in the in-house database were then analyzed using Compound Discoverer software (Thermo Fisher Scientific, Inc., Germany), where the threshold of MS/MS spectra similarity score was set as ≥ 40 with a maximum score of 100. The remaining peaks that did not have a match were then matched to the metabolites in our in-house MS database using the parent ion m/z and retention time. The thresholds for assignment using the parent ion m/z and retention time were ≤ 4 ppm and ≤ 0.15 min, respectively. Microbial metabolites were identified based on the MetAboliC pAthways DAtabase for Microbial taxonomic groups (MACADAM) and MetaCyc database[109]. In pathway analysis, metabolites with high ion intensity compared to the PBS control sample (enriched metabolites, fold change > 2) are used to find potential target pathways. We used the MetaboAnalyst[110] "Compound ID Conversion" tool to map metabolite names to KEGG database[111] IDs. Once we identify IDs, to examine the most up-to-date KEGG pathways, we collected KEGG pathways using KEGGREST R package[112]. Hypergeometric test $p$-values were computed to find KEGG pathways that are highly likely ($p$-value < 0.05) targeted by enriched metabolites in each sample.

## Analysis of metabolites with high-performance liquid chromatography (HPLC)
The fecal samples and bacteria growing medium were diluted with an equal volume of methanol. After centrifugation at $10,000 \times g$ for 30 min, 50 μl of supernatant was injected for high-performance liquid chromatography (HPLC) analysis. The HPLC analysis was performed on an Agilent 1260 Infinity system equipped with an Agilent ZORBAX SB-C18 column (4.6 × 150 mm, 3.5 μm), with following parameters: mobile phase A: 5 mM $NH_4Ac$ in water modified with 0.1% formic acid (v/v); mobile phase B: 5 mM $NH_4Ac$ in 90% acetonitrile modified with 0.1% formic acid (v/v); gradient: 5% B in first 5 min, 5–20% B for 10 min, hold 20% B for 5 min, 20–50% B for 5 min, hold 50% B for 5 min, 50–100% B for 5 min, hold 100% B for 10 min, 100–5% B for 5 min; flow rate: 1.0 ml/min; temperature: 30 °C. Peaks were identified by UV detection at 205 nm[113], and their identity was determined by comparing peak retention time with that of the standards. The HPLC standard for docosahexaenoic acid (DHA, Cat D2534) and eicosapentaenoic acid (EPA, Cat 44864) were purchased from Sigma for 4-oxo DHA (Cat 21373) was purchased from Cayman Chemical.

## RNA Extraction
Total RNA containing small RNA was isolated from PNP and murine tissues using a miRNeasy mini kit (Qiagen, Cat 217004) according to the manufacturer's instructions. In brief, 50 mg of PNP or tissue was disrupted in 100 μl of CTAB buffer (2% hexadecyl trimethyl ammonium bromide (CTAB), 2% polyvinylpyrrolidone (PVP), 2.0 M NaCl, 20 mM EDTA, 100 mM Tris-HCL (pH 8.0) and 5% β-mercaptoethanol) in a 60 °C water sonicate bath for 30 min. Tissue was homogenized using a tissue grinder before disruption. The homogenate was mixed with 700 μl of QIAzol Lysis Reagent and 140 μl of chloroform and centrifuged. The upper aqueous phase was mixed with 1.5 volumes of ethanol and loaded into an RNeasy spin column. The flow-through was discarded after centrifugation, and the column was washed with RWT and RPE sequentially. Total RNA was eluted with 50 μl RNase-free water. Bacterial mRNA was isolated using RiboPure Bacteria and MICROBExpress kits (Thermo Fisher Scientific, Cat AM1928 and AM1905) according to the manufacturers' instructions. The quality and quantity of the isolated RNA were analyzed using a NanoDrop spectrophotometer and Agilent Bioanalyzer.

## Quantitative real-time PCR for RNA expression
For analysis of gene mRNA expression, 1 μg of total RNA was reverse transcribed using SuperScript III reverse transcriptase (Invitrogen, Cat 12574026). Gene amplification and quantitation were performed using QuantiTect SYBR Green PCR Kit (Qiagen) and the listed primers (Supplementary Table 9). The quantitative PCR (qPCR) was performed using a BioRad CFX96 qPCR System with each reaction run in triplicate. Analysis and fold changes were determined using the comparative threshold cycle (Ct) method. Changes in mRNA expression was normalized by glyceraldehyde 3-phosphate dehydrogenase (GAPDH) and presented in the graph as fold-change.

## Bacterial gene knockout
The fusion PCR was used to knockout phospholipase (PLC) from *Clostridium perfringens* as previous described[48,49]. Two pairs of primers (Primer A/Fusion 1; Fusion 2/Primer D) were designed to amplify the upstream and downstream of the region to delete. The additional primer pair (AmpR) was used to amplify the antibiotic ampicillin resistance marker that will be used to replace the gene PLC. The sequences of the primers are listed (Supplementary Table 9). The final PCR product was electroporated directly into the bacteria in Tris-EDTA (TE, pH 8.0) buffer.

## Chromatin immunoprecipitation (ChIP) assay

The direct interaction of miR-159a and bacteria PLC DNA was identified with the ChIP assay. 1 nM of biotin-labeled miR-159-a (Eurofins) was incubated with B16F10 cells for 3 h at 37 °C following 37% formaldehyde (final 1%) in 1 ml PBS for one more hour for cross-linking. The cell lysate was collected in 1 ml of RIPA buffer (Sigma) and sonication was used to shear the chromatin into 0.5 ~ 1 kb fragment. Streptavidin Dynabeads (50 μl, Invitrogen, Cat 65305) were added to the DNA samples for 1 h at room temperature. PBS replaced the DNA and was incubated with the beads as the control to exclude non-specific interactions. After washing beads with PBS, biotinylated miR-159a complex was eluted from the beads with the biotin (4 mg/ml) in 25 mM Tris-HCL containing 0.3 M NaCl (pH 8.5) as the elution buffer at room temperature for 1 h or 95 °C for 5 min. The supernatant was collected and the DNA was purified with acetic acid (2%, v/v) and dissolved into 10 μl of $H_2O$. Two μl of DNA was used in qPCR reactions using the primers of PD-L1-ChIP (L1 (Supplementary Table 9)[114,115].

## Electron microscopy of isolated PNP

Isolated PNP were fixed in 2% paraformaldehyde (Electron Microscopy Science, PA) in PBS for 2 h at 22 °C followed by 1% glutaraldehyde (Electron Microscopy Science, PA) for 30 min at 22 °C. 15 μl of fixed samples were put on 2% agarose with formvar/carbon-coated nickel grids on top and allowed to absorb for 5–10 min. The grids with adherent exosomes were fixed in 2% paraformaldehyde in PBS for 10 min followed by extensive washing in PBS. Negative contrast staining was performed with 1.9% methyl cellulose and 0.3% uranyl acetate for 10 min. The grids with negatively stained exosomes were dried before observation under a Zeiss EM 900 electron microscope.

## Histological analysis

For hematoxylin and eosin (H&E) staining, tissues were fixed with buffered 10% formalin solution (SF93–20; Fisher Scientific, Fair Lawn, NJ) overnight at 4 °C[116]. Dehydration was achieved by sequential immersion in a graded ethanol series of 70, 80, 95, and 100% ethanol for 40 min each. Tissues were embedded in paraffin and subsequently cut into ultrathin slices (5 μm) using a microtome. Tissue sections were deparaffinized in xylene (Fisher), rehydrated in decreasing concentrations of ethanol in PBS. The hematoxylin and eosin were used to stain nuclear components and cytoplasmic components respectively. For tissue immunofluorescent staining, slides were washed three times (5 min each) with PBST. The tissue was permeabilized by incubating the slides in 1% Triton X-100 in PBS at 25 °C for 15 min and then washed three times in PBST. After blocking for 1 h at 25 °C in blocking buffer (PBS containing 10% bovine serum albumin (BSA)), slides were incubated overnight in a humidity chamber with anti-PD-L1 monoclonal antibody (BioCell, Cat BE0101). Antibodies were diluted 1:50 in the blocking buffer. Following another three PBST washes, slides were incubated with biotinylated anti-rat antibodies (Thermo Fisher, Cat 31830). Slides were then incubated with avidin-horseradish peroxidase (HRP) (Thermo Fisher, Cat 434423), followed by development with a DAB substrate kit (Thermo Fisher, Cat 34002). The light counterstaining with hematoxylin was used to reveal cells. The slides were mounted and scanned with an Aperio ScanScope.

## Thin-layer chromatography (TLC) analysis and lipid depletion

Lipids from PNP were extracted with chloroform/methanol and separated by TLC[18]. Briefly, HPTLC plates (silica gel 60 with a concentrating zone, 20 cm × 10 cm; Merck) were used for the separation. After aliquots of concentrated lipid samples, they were loaded on an HPTLC plate. The plate was placed vertically in a glass container, and the mixture of chloroform:methanol:water 65:25:4, v/v) was used as the mobile phase. The lipids were separated on the plate, and the bands of lipids on the plate were visualized in an iodine vapor chamber and imaged using an Odyssey Scanner (Licor Bioscience, Lincoln, NE). To

deplete a specific lipid, the lipid standard was loaded and migrated with the lipid sample in TLC. The band with the same travel distance as the standard lipid was removed. The residual lipid bands were collected for the lipid depletion study[4].

## Western blotting

For immunoblotting of tissue and cell line, lysates were prepared in modified radioimmunoprecipitation assay (RIPA) buffer (Sigma, Cat R0278) with the addition of protease and phosphatase inhibitors (Roche). Proteins were separated on 10% SDS-PAGE gels and transferred to PVDF membranes (BioRad Laboratories, Inc., Hercules, CA). Dual-color precision protein molecular weight (MW) markers (BioRad) were separated in parallel. Antibodies were purchased as follows: human PD-L1 (ab205921), mouse PD-L1 (ab213480), c-myc (ab32072), cleaved-caspase-3 (ab214430), cleaved-PARP (ab203467) and GAPDH (ab9485) antibodies from Abcam using a 1:1000 dilution. The secondary antibodies conjugated to Alex Fluor-647 was purchased from Invitrogen (Eugene, OR) using a 1:10,000 dilution. The bands were visualized using an Odyssey Imager (LiCor Inc, Lincoln, NE). For immunoblotting of bacterial proteins, the bacteria was harvested from 100–350 ml cultures by centrifugation at 5000 × g for 10 min. The cells were resuspended in 1 ml of TE buffer (100 mM Tris-Cl, 10 mM EDTA, pH 8.0) supplemented with protease inhibitor cocktail (Roche) and lysozyme (1 mg/mL) on ice for 45 min. The cell lysates were sonicated on ice using three 10 s bursts at medium intensity and then frozen in liquid nitrogen. The lysates were quickly thawed at 37 °C, and two more rapid sonication-freeze-thaw cycles were performed. Proteins were separated on 10% SDS-PAGE gels and transferred to PVDF membranes. Mouse anti-phospholipase C (PLC, *Clostridium perfringens*) antibody was purchased from Creative Diagnostics (CSC-H1598) and rabbit anti-*Clostridium perfringens* enterotoxin (CPE) antibody was purchased from GeneTex (GTX00879) using a 1:1000 dilution.

## Enzyme-linked immunosorbent assay (ELISA)

The level of PD-L1 in melanoma tissues of subjects was quantified using a Human PD-L1 SimpleStep ELISA kit (Abcam, Cat ab277712) according to the manufacturer's instructions. Briefly, 1 μg of tissue lysate in buffer PTR was incubated with 50 μl of antibody cocktail containing Capture antibody and Detector antibody in the microplate pre-coated with anti-PD-L1 at room temperature. After washing three times with buffer PT, the plate was incubated with TMB Development Solution for 10 min at 22 °C in the dark. The 100 μl of Stop Solution was added, and OD was recorded at 450 nm by a spectrometer (BioTek Synergy HT).

## Flow cytometry

Isolated cells were incubated with a blocking solution (0.5% BSA/PBS) for 15 min at 4 °C. For surface staining, cells were stained with various antibodies at room temperature for 1 h and then washed with PBS. Washed cells were stained for 40 min at 4 °C with the appropriate fluorochrome-conjugated antibodies in PBS with 2% FBS. Data were acquired using a BD FACSCalibur flow cytometer (BD Biosciences, San Jose, CA) and analyzed using FlowJo software (Tree Star Inc., Ashland, OR). The following antibodies purchased from BioLegend were used at a 1: 100 dilution for flow cytometry: anti-CD8 (100706), anti-PD-1 (109104), anti-IFNγ (505810), and FITC Annexin V (Cat 640945). Size gates set based on forward scatter (FSC) were applied in combination with quantification of fluorescence dye conjugated PNPs and antibodies.

## Single-strand gel shift assay

To identify the small molecule DHA binding site of the PD-L1 genome, we performed a single-strand gel shift assay as previously described[56]. Briefly, the synthesized DNA oligos (10 pmol) (Supplementary Table 10) were incubated with 1 nM of DHA in PBS/5% FBS at 37 °C for 30 min or 4 °C overnight following electrophoresis with 15% native

polyacrylamide gel electrophoresis (PAGE) gel without sodium dodecyl-sulfate (SDS) in Tris-acetate-EDTA (TAE) buffer. The oligos were stained with ethidium bromide (0.5 μg/ml) for 10 min and visualized under ultraviolet (UV) light.

## Surface plasmon resonance (SPR)

To identify the binding activity of metabolite DHA and PD-L1 promoter sequence, SPR experiments were conducted on an OpenSPR (Nicoya, Lifesciences, CA). Experiments were performed on a streptavidin sensor (Nicoya, Lifesciences). SPR was run at a flow rate of 20 μl/min using HBS running buffer (20 mM HEPES, 150 mM NaCl, pH 7.4). First, the streptavidin sensor chip was cleaned with octyl β-D-glucopyranoside (40 mM) and CHAPS (20 mM). 200 μl of biotin-conjugated oligos (1 μg/ml) with PD-L1 promoter sequence were injected on the sensor chip for 10 min until stable resonance was obtained. After immobilization of bio-oligo on the streptavidin sensor, the surface was blocked with BSA (3%) in a running buffer. After a stable signal was obtained, DHA (10 nM) was run over the immobilized bio-oligo until a stable resonance was obtained. A negative control test was also performed by injecting DHA onto a blank sensor chip to check for non-specific binding. The sensograms were analyzed using Trace-Drawer kinetic analysis software.

## Transient transfection and luciferase reporter assay

Murine melanoma B16F10 cells were plated in 24-well plates at a density of $3.0 \times 10^4$ cells/well in antibiotic-free DMEM medium supplemented with 10% FBS. 100 ng of luciferase reporter pEZX-mPDL1 or pEZX-hPDL1 (Genecopoeia) was transfected using Lipofectamine 3000 (Invitrogen, Cat L300000), P3000 reagent with Opti-MEM® Reduced Serum Medium (Thermo Fisher Scientific). For all reporter assays, the cell growth medium was harvested 48 hours post-transfection. The activities of Gaussia luciferase (GLuc) and secreted alkaline phosphatase (SEAP) in cell lysates were determined using the Secrete-Pair Dual Luminescence Assay Kit (Genecopoeia) and luminometer (BioTek). SEAP allows the normalization of the GLuc signal for endogenous control. Relative expression (fold-change) was determined by dividing the averaged normalized values from mock transfection. Values were averaged as indicated in the Figure legends.

## Quantification and statistical analysis

Unless otherwise indicated, all statistical analyses in this study were performed using SPSS 16.0 software. Data are presented as the mean ± standard deviation (SD). The significance of differences in mean values between the two groups was assessed using the Student's $t$ test. For comparisons involving more than two groups, differences between individual groups were analyzed using one-way or two-way ANOVA, followed by post-hoc t-tests. Differences in bacterial composition percentages were analyzed with a chi-square test. A $p$-value of less than 0.05 was considered statistically significant. Both animals and human subjects were randomly assigned to control and experimental groups matched for age and sex using simple randomization. Double-blinded procedures were used for animal and human studies. Unless otherwise indicated, the mice used in the in vivo study were male C57BL/6 J mice. Using one-way ANOVA to compare up to four groups, with a power of 0.7, a large effect size (0.75), and a significance level of 0.05, the minimum sample size needed in each group was 5[117]. The reported "n" represents the number of animals and human subjects in each study. Data are representative of at least three independent experiments.

## Reporting summary

Further information on research design is available in the Nature Portfolio Reporting Summary linked to this article.

## Data availability

All data generated or analyzed during this study are included in this published article and its Supplementary Information files or provided in a source data file. The microRNA and 16 s rDNA sequencing data were deposited in the National Center for Biotechnology Information (NCBI) Gene Expression Omnibus (GEO) database under the accession number GSE229897. (https://www.ncbi.nlm.nih.gov/geo/query/acc.cgi?acc=%20GSE229897). All data are included in the Supplementary Information or available from the authors, as are unique reagents used in this Article. The raw numbers for charts and graphs are available in the Source Data file whenever possible. Source data are provided in this paper.

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

## Acknowledgements

We thank Dr. J. Ainsworth for editorial assistance. This work was supported by a grant from the National Institutes of Health (NIH) (R01AT008617), the Robley Rex VA Medical Center Merit Review Grants (H.-G.Z.). Huang-Ge Zhang is supported by a Research Career Scientist (RCS) Award. X.Z. and M.M. are supported by P50 AA024337 and P20 GM113226. J.P. is supported by the NIH National Institute of General Medical Sciences (P20GM103436) and the NIH National Institute of Environmental Health Sciences grant (P30ES030283).

## Author contributions

Y.T. and H.G.Z. designed the study, analyzed and interpreted the data, and prepared the manuscript; C.L., X.Q., J.M., M.S., Q.X., M.L., F.X., L.Z., and R.X. performed the experiments and interpreted the data; J.P., J.H., M.K., Z.L., and X.H. performed bioinformatic analysis and X.Z., Y.J., M.M., and C.M. interpreted the findings.

## Competing interests

The authors declare no competing interests.
