## [Transparent Peer Review file · Nature Communications]

Plant-nanoparticles enhance anti-PD-L1 efficacy by shaping human commensal microbiota metabolites

Corresponding Author: Professor Huang-Ge Zhang

Version 0:

Reviewer comments:

Reviewer #1

(Remarks to the Author)

Yun et al. demonstrated that the plant-nanoparticles (PNPs) were selectively taken up by gut bacteria post gavage through cell signals. Through lipidomic analysis and protein/amino acid profiling, it was determined that ginger-derived exosome-like nanoparticles (GELNs) contained a higher level of digalactosyldiacylglycerol (DGDG) and enriched glycine, which function as "Eat me" or "Don't eat me" signals for uptake. Due to these unique properties, GELNs are preferentially taken up by a specific family of human gut bacteria, which can eventually enhance anti-PD-L1 efficacy in melanoma by reprogramming bacterial unsaturated fatty acid metabolism. Furthermore, to identify the underlying mechanism of GELNs in the production of DHA and EPA, the authors performed NGS miRNA sequencing and provided abundant miRNA sequences of GELN-RNAs. Based on the genome database, phospholipase C (PLC) was identified as a potential target of GELN miRNA. Overall, the research team provided extensive insights into the gut microbiome and offered a strategy using GELNs to enhance tumor immunotherapy efficacy through the molecular regulation of DHA and EPA. Generally, the study was well organized and the results were solid and sufficient. However, some concerns should be addressed before the acceptance.

Major concerns:

1. In the study, GELN was orally administered, but there was no evidence of its stability in vivo. As the author investigated the composition of GELN, GELN consisted of a large amount of proteins and lipids, which can be disrupted in the gastrointestinal tract. Is GELN stable in the harsh conditions in vivo, both physically and chemically? Aren't lipids and protein/amino acids destroyed or damaged in the gastrointestinal tract?
2. Why was GELN specifically selected as a representative ELN for anti-PD-L1 therapy? Is there any particular reason for choosing GELN over other ELNs?
3. According to the lipid profiles of ELNs, digalactosyldiacylglycerol (DGDG) is comparably enriched in various ELNs, such as those derived from notoginseng and turmeric. If the high level of DGDG is considered very important in this system, shouldn't an appropriate control that enriches DGDG, like GELN, have been included?
4. According to the heatmap in Figure 2F, the correlation of lipids in GELN versus bacteria showed a comparably higher correlation between MGDG and Lactobacillaceae than between DGDG and Lactobacillaceae. However, the article insisted that GELN is preferentially taken up by Lachnospiraceae and Lactobacillaceae, mediated by digalactosyldiacylglycerol (DGDG). Is there a discrepancy between the correlation data and the actual experimental results?
5. I understood that the administration of GELN can successfully induce EPA and DHA in vivo, which are key molecular mechanisms enhancing the anti-PD-L1 immunotherapy in the study. I wondered how the research team specifically determined that GELN aly-miR159a-3p can regulate the metabolism of EPA and DHA during the screening step and identified phospholipase C (PLC) as a potential target?
6. How do we define PNPs and ELNs compared to mammalian-cell-derived exosomes? Is there a particular standard or evidence to distinguish between PNPs and mammalian-cell-derived exosomes?
7. In line 190, feces were isolated from mice 2 h after gavage, how is this time point chosen? Is 2 hours enough for PNPs to be taken up by gut bacteria? Further, whether the PNPs still intact should also be evaluated.
8. In line 199-200, the authors demonstrated that GELN and ginger Nano10 were stable in gastric juice. However, gastric juice is acidic and contains many types of proteases, what's the mechanism that GELNs keep stable in it? Further, only size and concentration were not enough to demonstrate the stability of GELNs, it would be better if the authors can further investigate the intactness of its cargo protein and nucleic acids and whether GELN could exert its original biological

function.

9. In the section named "GELN RNAs modulate gut bacterial metabolite activity by contributing to the inhibition of melanoma growth and metastasis in a mouse model", authors only investigated the effect of RNAs in immune-therapy, however, the effect of lipids and protein component should also be evaluated.

10. The researchers demonstrated aly-miR159a-3p could target bacterial phospholipase C, which leads to DHA accumulation and eventually enhanced anti-tumor effect of PD-L1 therapy. Was it sufficient for aly-miR159a-3p from GELN to exert the effect? since to our knowledge, the concentration of miRNA inside PNP is very low.

Minor concerns:

1. In Figure 1E, the percent sign '%' is repeated in the graph. Please revise the figure.

2. In Figure 3A, when treating PNPs in mice, 0.5g/kg PNPs was used. Is this based on the protein concentration or the weight of PNPs? Please clarify.

3. In Sup. Figure 3, Figure 6E, there are different notation styles for time in the manuscript. For example, 1 h and 3hs. Please unify notation style and fix the spacing after numbers.

4. In Fig. 4 B, C and D, Figure 6A and N, usually at least 6 tumor tissues should be displayed for each group.

5. In Figure 6I, western blotting image of GAPDH displayed different band intensities for each sample. It's hard to analyze whether the expression of PD-L1 in the samples has increased or decreased. And full image for western blot results should be provided, for Fig. 6D and 6I.

6. In Figure 7B, the lengths of 60 nt of oligo PD-L1 were visualized in a 15% PAGE gel after treatment with DHA. Although the same length oligos of PD-L1p8, p9, and p10 were loaded on the native page gel, oligo PD-L1p9 was found slightly towards the top of the gel, while those of PD-L1p8 and p10 were found towards the bottom of the gel. As I understand, nucleotides are separated by their molecular weight/length of oligo in page gel electrophoresis. However, based on the result of gel electrophoresis, it seems that the length of oligos of PD-L1p8, p9, and p10 is different. Why did the same-sized DNA fragments of PD-L1p8, p9, and p10 migrate to different locations?

Reviewer #2

(Remarks to the Author)

In this study, Teng et al investigated the effects of plant-nanoparticles on anti-PD-L1 efficacy and the mechanisms involved by using a humanized gnotobiotic mouse model. They showed nicely that plant-derived nanoparticles were taken up by human gut bacteria which was mediated by the lipids and amino acids, leading to altered gut bacterial metabolites with an accumulation of docosahexaenoic acid (DHA). They further showed that ginger-derived exosome-like nanoparticles (GELN) preferentially uptake by Lachnospiraceae and Lactobacillaceae through different pathways. Increased circulating DHA epigenetically regulates PD-L1 expression in tumor cells by binding the PD-L1 promoter and subsequently preventing c-myc-initiated transcription of PD-L1. Importantly, colonization of germ-free mice with gut bacteria from anti-PD-L1 non-responding patients supplemented with DHA significantly enhanced the efficacy of anti-PD-L1 therapy. This is an interesting study with comprehensive approaches, including multi-omics and an interesting animal model. The conclusion is supported by solid data. I only have a few minor concerns that need to be addressed to benefit the readers:

1) There are a few grammars need to be corrected.

2) Fig 4D, it is unclear why the GNV-RNAs treatment did not affect tumor development while GNV-RNAs SuP treatment inhibited tumor size as GNV-RNAs treatment should also alter the gut microbiota metabolites similar to that in the Sup. Please discuss.

Version 1:

Reviewer comments:

Reviewer #1

(Remarks to the Author)

The authors have addressed all the points raised in my previous review and provided satisfactory responses. The additional experimental results are convincing and strengthen the study's findings. The manuscript's clarity and language have been improved, making it more readable and professional. I am pleased with the revisions and am willing to accept the manuscript for publication after any remaining minor language issues are addressed.

Reviewer #2

(Remarks to the Author)

All my previous concerns have been addressed appropriately.

October 21, 2024

RE: Manuscript ID: NCOMMS-24-37764-T

Title: Plant-nanoparticles enhance anti-PD-L1 efficacy by shaping human commensal microbiota

REVIEWER COMMENTS

Reviewer #1 (Remarks to the Author):

Yun et al. demonstrated that the plant-nanoparticles (PNPs) were selectively taken up by gut bacteria post gavage through cell signals. Through lipidomic analysis and protein/amino acid profiling, it was determined that ginger-derived exosome-like nanoparticles (GELNs) contained a higher level of digalactosyldiacylglycerol (DGDG) and enriched glycine, which function as "Eat me" or "Don't eat me" signals for uptake. Due to these unique properties, GELNs are preferentially taken up by a specific family of human gut bacteria, which can eventually enhance anti-PD-L1 efficacy in melanoma by reprogramming bacterial unsaturated fatty acid metabolism. Furthermore, to identify the underlying mechanism of GELNs in the production of DHA and EPA, the authors performed NGS miRNA sequencing and provided abundant miRNA sequences of GELN-RNAs. Based on the genome database, phospholipase C (PLC) was identified as a potential target of GELN miRNA. Overall, the research team provided extensive insights into the gut microbiome and offered a strategy using GELNs to enhance tumor immunotherapy efficacy through the molecular regulation of DHA and EPA. Generally, the study was well organized and the results were solid and sufficient. However, some concerns should be addressed before the acceptance.

Major concerns:

1. In the study, GELN was orally administered, but there was no evidence of its stability in vivo. As the author investigated the composition of GELN, GELN consisted of a large amount of proteins and lipids, which can be disrupted in the gastrointestinal tract. Is GELN stable in the harsh conditions in vivo, both physically and chemically? Aren't lipids and protein/amino acids destroyed or damaged in the gastrointestinal tract?

Response: Like exosomes which are stable in the gut¹, plant-derived exosomes-like nanoparticles (ELNs) have similar components as mammalian exosomes and are stable in vivo via oral administration²⁻⁵. The results published from our group and others indicate that ELNs including Ginger ELN (GELN) show a high tolerance to the gastrointestinal acidic environment, enzymes and bile extracts²⁻⁵. In the revised manuscript, using GELN as an example, we tested GELN stability under the gastric condition for 3 h and GELN cargo such as proteins and RNA are still intact (**Sup. Fig 3F**), suggesting that the PNPs are stable under gastric conditions.

Besides ELNs, numerous nanoparticles including exosomes which consist of protein, RNA and lipids can be detected in the gut and are protected from being destroyed. In the discussion section (page 24, lines 826-834) of the revised manuscript, we further discussed the possible mechanism underlying the protections.

2. Why was GELN specifically selected as a representative ELN for anti-PD-L1 therapy? Is there any particular reason for choosing GELN over other ELNs?

Response: In the revised manuscript, we demonstrated that GELN significantly enhances anti-PD-L1 therapy (Figure 4B), and ginger-derived nano10, garlic-ELN, aloe-ELN and lemon-ELN did not enhance anti-PD-L1 therapy in a mouse melanoma model (**Sup. Fig. 7A**).

3. According to the lipid profiles of ELNs, digalactosyldiacylglycerol (DGDG) is comparably enriched in various ELNs, such as those derived from notoginseng and turmeric. If the high level of DGDG is considered very important in this system, shouldn't an appropriate control that enriches DGDG, like GELN, have been included?

Response: We appreciate this reviewer's suggestion. The role of ELN DGDG is to provide a "eat me" signal. In the revised manuscript, the uptake of GELN by *C. perf* was included in the analysis of DGDG mediated *C. perf* uptake (Sup. Fig. 5D, top panel). To test the effect of GELN DGDG on bacterial uptake, the total lipids were extracted from GELN and assembled into nanovesicles (GNV) which consisted of the same lipid profile as GELNs. The results suggested that GELN and GNV have similar uptake efficiencies. The depletion of DGDG in GNV causes a reduction of uptake by *C. perf*. We further tested whether other ELN lipids influence the DGDG enriched ELNs uptake. Aloe-ELN contains high levels of DGDG (Figure 2A) so we used aloe ELN as an example. The result suggests that aloe-ELN displays a low uptake by *L. reuteri* (Sup. Fig. 5D, bottom panel), due to aloe-ELN being highly rich with PI which inhibits the uptake of the bacteria. The depletion of aloe-ELN PI causes an increase of uptake by *L. reuteri*. Therefore, even though the "eat me" signal is presented on the ELNs, the efficiency of up take is dependent upon whether a "do not eat me" signal is present on the ELNs as well. We further discussed other ELN factors that could affect the uptake in the revised manuscript, discussion section, page 24, paragraphs 2-5.

4. According to the heatmap in Figure 2F, the correlation of lipids in GELN versus bacteria showed a comparably higher correlation between MGDG and Lactobacillaceae than between DGDG and Lactobacillaceae. However, the article insisted that GELN is preferentially taken up by Lachnospiraceae and Lactobacillaceae, mediated by digalactosyldiacylglycerol (DGDG). Is there a discrepancy between the correlation data and the actual experimental results?

Response: Overall, in this study, we tested whether there are "eat me" signals and "do not eat me" signals presented on the PNPs. From correlation analyses, we identified the top 10 lipid (Fig. 2G) and top 10 amino acids (Fig. 2H) that can serve as signals for regulation of gut bacteria uptake. Among these 10 lipids, 7 lipids including DGDG and MGDG could serve as "eat me" signals and 3 lipids as "do not eat me" signals. We randomly selected DGDG which is one of these 7 candidates to experimentally test whether it play a causative role in up taken by gut bacteria. In brief, to experimentally verify our analysis of the correlation, suggesting that lipids of PNPs can enhance/inhibit uptake by gut bacteria, we conducted the uptake tests with the specific species of bacteria. Because of the commercial availability of specific species of bacteria, we used *Clostridium perfringens* (*C. perf*), a species in the Lachnospiraceae family and *Lactobacillus reuteri* (*L. reuteri*), a species in the Lactobacillaceae family to test our lipids correlation results. We randomly found that DGDG ("Eat me") and PI ("Do not eat me") from the Lachnospiraceae family preferentially take up DGDG enriched GELN (Fig. 1B, Sup. Table 1C-1E) (Figs. 2A and 2G), and PI enriched AELN inhibits Lactobacillaceae uptake of AELN (Figs. 2F and 2G). The DGDG and PI were depleted from the total lipids of GELN and AELN, respectively⁴ and the remainder of the lipids were reassembled into a nano-sized nanovesicle (NV) by sonication with the protocol we published⁴, and referred to them as ginger NV (GNV) and aloe NV (ANV), respectively. The FACS analysis indicated that the depletion of DGDG from the GELN lipids significantly reduced *C. perf* uptake of GNV whereas the addition of DGDG rescued the uptake of GNV (Sup. Fig. 5D, top panel). In contrast, the depletion of PI promotes the uptake of ANV by *L. reuteri* and the addition of PI restored the inhibitory effect on the uptake of ANV (Sup. Fig. 5D, bottom panel).

5. I understood that the administration of GELN can successfully induce EPA and DHA in vivo, which are key molecular mechanisms enhancing the anti-PD-L1 immunotherapy in the study. I wondered

how the research team specifically determined that GELN aly-miR159a-3p can regulate the metabolism of EPA and DHA during the screening step and identified phospholipase C (PLC) as a potential target?

Response: In the revised manuscript, we further explain in detail (page 14, lines 476-491) that all high abundance miRNAs sequences identified in the GELN (Fig. 5C) were input into the bacterial genomes database from the NCBI RefSeq database to search for potential targets of GELN miRNAs. An alignment of nucleotides sequences using BLAST indicated that only GELN miRNA aly-miR159a-3p potentially targets phospholipase C (PLC) with two 7-mer lengths of reverse complement sequences (Fig. 5D), which contain a lipoxygenase domain that may catalyze DHA. Given GELN and not other PNPs can enhance immunotherapy efficacy in melanoma and specifically raise DHA production in the gut (Figs. 3A and 5A), we hypothesized that GELN enhances the DHA level by miR159a targeting bacterial PLC, resulting in a positive promotion of immunotherapy in melanoma.

6. How do we define PNPs and ELNs compared to mammalian-cell-derived exosomes? Is there a particular standard or evidence to distinguish between PNPs and mammalian-cell-derived exosomes?

Response: In the revised manuscript, we further defined PNPs and ELNs compared to mammalian-cell-derived exosomes in the discussion section (page 24, paragraph 4). In brief, plant-derived nanoparticles (PNPs) are referred to as nanoparticles isolated from edible plants. As a proof of concept, Nano10 and ELNs were isolated and studied. Nano10 is derived from the pellet after a 10,000xg centrifugation, while ELNs are derived from the pellet after a 100,000xg centrifugation. Unlike mammalian-cell-derived exosomes, which are spontaneously released from mammalian cells, Nano10 and ELNs are not spontaneously released from plant cells. Instead, they are released when the edible plant cell is broken down, such as during chewing and digestion. On the other hand, ELNs have similar components, including proteins, lipids, and RNA, as mammalian cell exosomes. We expect that, in addition to Nano10 and ELNs, other types of PNPs are present in the plant diet. Currently, there is no standard markers to distinguish between ELNs and mammalian-cell-derived exosomes.

7. In line 190, feces were isolated from mice 2 h after gavage, how is this time point chosen? Is 2 hours enough for PNPs to be taken up by gut bacteria? Further, whether the PNPs still intact should also be evaluated.

Response: Food can reach the large intestine as early as one hour after being administered via oral gavage⁶. In the revised manuscript, we provide the rationale for choosing this time point in the result section (page 5, line 179-189) and the data show that bacteria do take up the PNPs in vitro within 30 minutes (Sup. Fig. 3A, right panel). After incubating GELNs with gastric fluid at 37 °C for 3 hours the cargo protein and RNA are still intact (Sup. Fig. 3F). Therefore, 2h after gavage was chosen for identifying the specific bacteria taking up the fluorescent dye-labelled PNPs.

8. In line 199-200, the authors demonstrated that GELN and ginger Nano10 were stable in gastric juice. However, gastric juice is acidic and contains many types of proteases, what's the mechanism that GELNs keep stable in it? Further, only size and concentration were not enough to demonstrate the stability of GELNs, it would be better if the authors can further investigate the intactness of its cargo protein and nucleic acids and whether GELN could exert its original biological function.

Response: We appreciate this suggestion. In the revised manuscript, we analyzed the stability of the GELN cargo protein and RNA under gastric condition. The result indicated that, beside size and concentration, the GELN cargo protein and RNA (Sup. Fig. 3F) are also intact. More importantly, our data suggests that administration via oral gavage of GELN or GNV-RNA made from lipid extracted from GELN and containing GELN derived RNA exerts a similar efficacy in enhancing anti-PD-L1 therapy in a melanoma mouse model (Figs. 4B and 4C).

9. In the section named “GELN RNAs modulate gut bacterial metabolite activity by contributing to the inhibition of melanoma growth and metastasis in a mouse model”, authors only investigated the effect of RNAs in immune-therapy, however, the effect of lipids and protein component should also be evaluated.

Response: In the revised manuscript, in Fig. 4, we identified the impact of the contents in GELN. We generated nanovesicles (GNV) made from total lipids extracted from GELN, which contain only lipids without GELN proteins, metabolites and nucleic acids included. The GELN-RNA was extracted from GELN and encapsulated back into GNV (GNV-RNA). We found that GNV-RNA without GELN protein can significantly enhance the anti-PD-L1 efficacy, indicating that GELN RNA play a causative role in the enhancing of the immunotherapeutic effect (Fig. 4C). To exclude the effect of GELN lipids, we used non-specific scramble miRNA encapsulated into GNV as a control, the result shows that GNV-scramble miRNA has no benefit on anti-PD-L1 therapy efficacy.

10. The researchers demonstrated aly-miR159a-3p could target bacterial phospholipase C, which leads to DHA accumulation and eventually enhanced anti-tumor effect of PD-L1 therapy. Was it sufficient for aly-miR159a-3p from GELN to exert the effect? since to our knowledge, the concentration of miRNA inside PNP is very low.

Response: The yield of PNPs (Sup. Fig. 1I) indicates that each kg of plant tissue can generate 1-5 g of PNPs and that one to five milligrams of RNA is isolated from each gram of ELN⁴. Aly-miR159s-3p is one of the richest miRNAs in GELN (Fig. 5C). It has been established in vivo that extracellular vesicles including exosomes exert apparent biological functions mediated by cargo miRNAs^{7, 8}, which is providing a therapeutic potential for diseases. In our experience, the low yield of miRNA isolated from plant-derived PNPs is usually attributed to the incomplete lysate of these NPs. Regular lysate buffer such as RIPA does not work well to lyse these plant-nanoparticles. We had a problem isolating RNA from plant or plant-derived NPs until we used modified CTAB buffer in combination with sonication before mixing with Trizol. The detail method is provided in the Methods section, RNA Extraction (page 30, lines 1056-1069).

Minor concerns:

1. In Figure 1E, the percent sign ‘%’ is repeated in the graph. Please revise the figure.

Response: We corrected the mistake in the revised manuscript.

2. In Figure 3A, when treating PNPs in mice, 0.5g/kg PNPs was used. Is this based on the protein concentration or the weight of PNPs? Please clarify.

Response: It is based on the weight of PNPs. we clarified this point in page 10, line 364.

3. In Sup. Figure 3, Figure 6E, there are different notation styles for time in the manuscript. For example, 1 h and 3hs. Please unify notation style and fix the spacing after numbers.

Response: This issue has been corrected and all styles of time have been unified.

4. In Fig.4 B, C and D, Figure 6A and N, usually at least 6 tumor tissues should be displayed for each group.

Response: We used five mice in each group of treatment. The representative tumor or tissue in each group was presented in the figure due to the space limitations and the plots are used to show the average tumor volume and lung metastasis nodules.

5. In Figure 6I, western blotting image of GAPDH displayed different band intensities for each sample. It's hard to analyze whether the expression of PD-L1 in the samples has increased or decreased. And full image for western blot results should be provided, for Fig. 6D and 6I.

Response: The intensity of each band was quantified using ImageJ. Numbers below western blots represent densitometry values normalized to the loading control. We explained this in the Fig. 6 legend. All western blot uncropped images are provided in the Source data file.

6. In Figure 7B, the lengths of 60 nt of oligo PD-L1 were visualized in a 15% PAGE gel after treatment with DHA. Although the same length oligos of PD-L1p8, p9, and p10 were loaded on the native page gel, oligo PD-L1p9 was found slightly towards the top of the gel, while those of PD-L1p8 and p10 were found towards the bottom of the gel. As I understand, nucleotides are separated by their molecular weight/length of oligo in page gel electrophoresis. However, based on the result of gel electrophoresis, it seems that the length of oligos of PD-L1p8, p9, and p10 is different. Why did the same-sized DNA fragments of PD-L1p8, p9, and p10 migrate to different locations?

Response: The double strand nucleotides are separated by their molecular length in PAGE. However, single strand nucleotides are separated by conformation^{9, 10}. The conformation of nucleotides in PAGE affects how they move through the gel, and this can be used to study DNA integrity and mutation. We developed a protocol to determine whether small molecules bind to a single strand oligo nucleic acid¹¹. Considering that the single strand nucleic acid chain exhibits conformational differences on non-denaturing PAGE¹², we assumed that single-strand DNA will display an electrophoretic mobility shift on native PAGE when the small molecules bind to DNA, which is where the named single-strand gel shift (SSGS) is derived. The oligos we tested contained different sequences and exhibited different migration in non-denaturing PAGE although they are the same length. Also, the migration will shift if the oligo is bound by a small molecule in comparison with the oligo without a small molecule being bound.

Reviewer #2 (Remarks to the Author):

In this study, Teng et al investigated the effects of plant-nanoparticles on anti-PD-L1 efficacy and the mechanisms involved by using a humanized gnotobiotic mouse model. They showed nicely that plant-derived nanoparticles were taken up by human gut bacteria which was mediated by the lipids and amino acids, leading to altered gut bacterial metabolites with an accumulation of docosahexaenoic acid (DHA). They further showed that ginger-derived exosome-like nanoparticles (GELN) preferentially uptake by Lachnospiraceae and Lactobacillaceae through different pathways. Increased circulating DHA epigenetically regulates PD-L1 expression in tumor cells by binding the PD-L1 promoter and subsequently preventing c-myc-initiated transcription of PD-L1. Importantly, colonization of germ-free mice with gut bacteria from anti-PD-L1 non-responding patients supplemented with DHA significantly enhanced the efficacy of anti-PD-L1 therapy. This is an interesting study with comprehensive approaches, including multi-omics and an interesting animal model. The conclusion is supported by solid data. I only have a few minor concerns that need to be addressed to benefit the readers:

1) There are a few grammars need to be corrected.

Response: The revised manuscript has been corrected by people whose native language is English.

2) Fig 4D, it is unclear why the GNV-RNAs treatment did not affect tumor development while GNV-RNAs SuP treatment inhibited tumor size as GNV-RNAs treatment should also alter the gut microbiota metabolites similar to that in the Sup. Please discuss.

Response: We used germ free mice (GF) to identify whether gut bacteria contribute to the influence of PNPs on tumor immunotherapy. GNV-RNA treatment did not enhance the immunotherapy in GF mice while GNV-RNAs Sup treatment enhance the immunotherapy (Figure 4D). The GNV-RNAs Sup is the

gut fecal supernatant collected from the normal SPF mice treated with GNV-RNAs (page 14, line 453-455), which contain gut bacterial metabolites. This data suggests that GNV-RNAs shape the gut bacterial metabolites contributing to the enhancement of the anti-PD-1L immunotherapy. We corrected the description (page 14, line 453).

References:

1. Smythies LE, Smythies JR. Exosomes in the gut. *Front Immunol.* 2014;5:104. Epub 20140317. doi: 10.3389/fimmu.2014.00104. PubMed PMID: 24672525; PMCID: PMC3955839.
2. Wang B, Zhuang X, Deng ZB, Jiang H, Mu J, Wang Q, Xiang X, Guo H, Zhang L, Dryden G, Yan J, Miller D, Zhang HG. Targeted drug delivery to intestinal macrophages by bioactive nanovesicles released from grapefruit. *Mol Ther.* 2014;22(3):522-34. Epub 20130813. doi: 10.1038/mt.2013.190. PubMed PMID: 23939022; PMCID: PMC3944329.
3. Subudhi PD, Bihari C, Sarin SK, Baweja S. Emerging Role of Edible Exosomes-Like Nanoparticles (ELNs) as Hepatoprotective Agents. *Nanotheranostics.* 2022;6(4):365-75. Epub 20220621. doi: 10.7150/ntno.70999. PubMed PMID: 35795340; PMCID: PMC9254361.
4. Teng Y, Ren Y, Sayed M, Hu X, Lei C, Kumar A, Hutchins E, Mu J, Deng Z, Luo C, Sundaram K, Sriwastva MK, Zhang L, Hsieh M, Reiman R, Haribabu B, Yan J, Jala VR, Miller DM, Van Keuren-Jensen K, Merchant ML, McClain CJ, Park JW, Egilmez NK, Zhang HG. Plant-Derived Exosomal MicroRNAs Shape the Gut Microbiota. *Cell Host Microbe.* 2018;24(5):637-52.e8. Epub 20181025. doi: 10.1016/j.chom.2018.10.001. PubMed PMID: 30449315; PMCID: PMC6746408.
5. Zhuang X, Deng ZB, Mu J, Zhang L, Yan J, Miller D, Feng W, McClain CJ, Zhang HG. Ginger-derived nanoparticles protect against alcohol-induced liver damage. *J Extracell Vesicles.* 2015;4:28713. Epub 20151125. doi: 10.3402/jev.v4.28713. PubMed PMID: 26610593; PMCID: PMC4662062.
6. Padmanabhan P, Grosse J, Asad AB, Radda GK, Golay X. Gastrointestinal transit measurements in mice with 99mTc-DTPA-labeled activated charcoal using NanoSPECT-CT. *EJNMMI Res.* 2013;3(1):60. Epub 20130802. doi: 10.1186/2191-219x-3-60. PubMed PMID: 23915679; PMCID: PMC3737085.
7. Xu D, Di K, Fan B, Wu J, Gu X, Sun Y, Khan A, Li P, Li Z. MicroRNAs in extracellular vesicles: Sorting mechanisms, diagnostic value, isolation, and detection technology. *Front Bioeng Biotechnol.* 2022;10:948959. Epub 20221017. doi: 10.3389/fbioe.2022.948959. PubMed PMID: 36324901; PMCID: PMC9618890.
8. López de Las Hazas MC, Tomé-Carneiro J, Del Pozo-Acebo L, Del Saz-Lara A, Chapado LA, Balaguer L, Rojo E, Espín JC, Crespo C, Moreno DA, García-Viguera C, Ordoñas JM, Visioli F, Dávalos A. Therapeutic potential of plant-derived extracellular vesicles as nanocarriers for exogenous miRNAs. *Pharmacol Res.* 2023;198:106999. Epub 20231119. doi: 10.1016/j.phrs.2023.106999. PubMed PMID: 37984504.
9. Celotto AM, Graveley BR. Using single-strand conformational polymorphism gel electrophoresis to analyze mutually exclusive alternative splicing. *Methods Mol Biol.* 2004;257:65-74. doi: 10.1385/1-59259-750-5:065. PubMed PMID: 14769996; PMCID: PMC2376758.
10. Pasookhush P, Usmani A, Suwannahong K, Palittapongarnpim P, Rukseree K, Ariyachaokun K, Buates S, Siripattanapipong S, Ajawatanawong P. Single-Strand Conformation Polymorphism Fingerprint Method for Dictyostelids. *Front Microbiol.* 2021;12:708685. Epub 20210827. doi: 10.3389/fmicb.2021.708685. PubMed PMID: 34512585; PMCID: PMC8431811.
11. Teng Y, Mu J, Xu F, Zhang X, Sriwastva MK, Liu QM, Li X, Lei C, Sundaram K, Hu X, Zhang L, Park JW, Hwang JY, Rouchka EC, Zhang X, Yan J, Merchant ML, Zhang HG. Gut bacterial isoamylamine promotes age-related cognitive dysfunction by promoting microglial cell death. *Cell Host Microbe.* 2022;30(7):944-60.e8. Epub 2022/06/03. doi: 10.1016/j.chom.2022.05.005. PubMed PMID: 35654045; PMCID: PMC9283381.

12. Liu Q, Scaringe WA, Sommer SS. Discrete mobility of single-stranded DNA in non-denaturing gel electrophoresis. *Nucleic Acids Res.* 2000;28(4):940-3. Epub 2000/01/29. doi: 10.1093/nar/28.4.940. PubMed PMID: 10648786; PMCID: PMC102567.